# Efficient and Flexible Neural Network Training through Layer-wise Feedback Propagation

**Leander Weber**[1]    **Jim Berend**[1]    **Moritz Weckbecker**[1]    **Alexander Binder**[2,3]    **Thomas Wiegand**[1,4,5]
**Wojciech Samek**[1,4,5,†]    **Sebastian Lapuschkin**[1,6,†]

[1] *Fraunhofer Heinrich Hertz Institute, Berlin, Germany*

[2] *Otto-von-Guericke University, Magdeburg, Germany*

[3] *Singapore Institute of Technology, Singapore, Singapore*

[4] *Technische Universität Berlin, Berlin, Germany*

[5] *BIFOLD – Berlin Institute for the Foundations of Learning and Data, Berlin, Germany*

[6] *Centre of eXplainable Artificial Intelligence, Technological University Dublin, Dublin, Ireland*

[†] *corresp.:* `{wojciech.samek,sebastian.lapuschkin}@hhi.fraunhofer.de`

**Reviewed on OpenReview:** *https://openreview.net/forum?id=9oToxYVOSW*

## Abstract

Gradient-based optimization has been a cornerstone of machine learning that enabled the vast advances of Artificial Intelligence (AI) development over the past decades. However, this type of optimization requires differentiation, and with recent evidence of the benefits of non-differentiable (e.g. neuromorphic) architectures over classical models w.r.t. efficiency, such constraints can become limiting in the future. We present Layer-wise Feedback Propagation (LFP), a novel training principle for neural network-like predictors that utilizes methods from the domain of explainability to decompose a reward to individual neurons based on their respective contributions. Leveraging these neuron-wise rewards, our method then implements a greedy approach reinforcing helpful parts of the network and weakening harmful ones. While having comparable computational complexity to gradient descent, LFP does not require gradient computation and generates sparse and thereby memory- and energy-efficient parameter updates and models. We establish the convergence of LFP theoretically and empirically, demonstrating its effectiveness on various models and datasets. Via two applications — neural network pruning and the *approximation-free* training of Spiking Neural Networks (SNNs) — we demonstrate that LFP combines increased efficiency in terms of computation and representation with flexibility w.r.t. choice of model architecture and objective function. Our code is available at `https://github.com/leanderweber/layerwise-feedback-propagation`.

## 1 Motivation

In recent years, supervised deep learning has been successfully employed in a wide variety of applications, including highly accurate classification (Dosovitskiy et al., 2021), generation of realistic images (Rombach et al., 2022), writing of complex text (Ouyang et al., 2022), or assisting in medical diagnosis (Briganti & Le Moine, 2020). These developments have been enabled by the widespread use of well-researched gradient-based training methods that provide fast convergence and are easily implemented on GPU hardware. However, the computation of gradients requires the model and objective to be differentiable, which can pose a significant restriction and limit flexibility in terms of objective formulation and model architecture: For instance, gradient-based training becomes difficult when non-differentiable evaluation measures should be integrated into the objective function (e.g. F1-Score in classification or Levenshtein distance in word recognition tasks) or when feedback is obtained from external sources such as humans (Christiano et al., 2017). Similarly, gradients cannot be used to directly train non-differentiable models despite these models providing unique benefits, especially when implemented on dedicated hardware; e.g., when a model is quantized to have discrete forward passes for reduced memory requirements or when neuromorphic architectures such as Spiking Neural

## Layer-wise Feedback Propagation

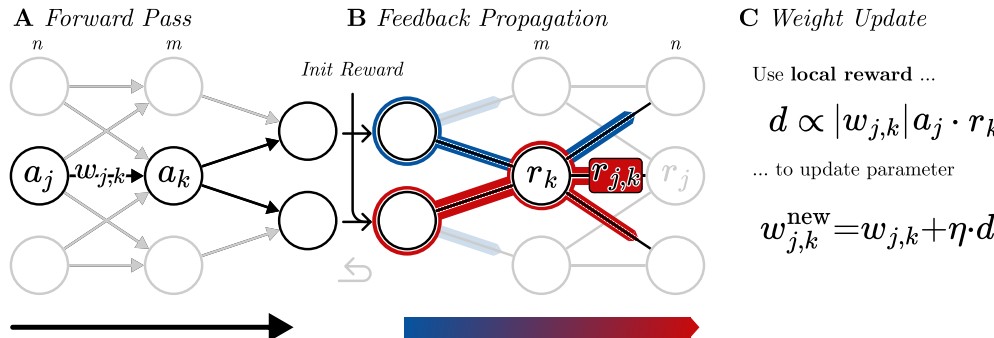

Figure 1: Our proposed method. *A*: During the forward pass, the model receives an input and generates activations $a_j, a_k$ and an output. *B*: After receiving an *initial reward* $r_c$ (cf. Sections 3.1.1, 3.2) that evaluates the output quality, Layer-wise Feedback Propagation (LFP) decomposes $r_c$ into *local rewards* $r_k$ for each neuron $k$ (and rewards $r_{jk}$ for each weighted connection $j \to k$). This is achieved via a backward pass that leverages Layer-wise Relevance Propagation (LRP)-rules and does not require gradient computation, obtaining feedback for each neuron that evaluates how well it performed w.r.t. a given task. For this step, LFP requires an initial reward for each output neuron, but otherwise places no restrictions on how this reward is obtained. *C*: After the initial reward is decomposed into local rewards, each neuron is viewed as a separate agent with its own (local) reward that is used to derive a parameter update $d_{w_{jk}}^{\text{lfp}}$. This update implements a greedy strategy, strengthening connections that are evaluated as helpful by receiving a positive reward, and weakening connections that receive a negative reward. Note that the Hebbian-like update in step C could be performed based on any local reward, and is not necessarily tied to the specific propagation in step B we employ in this work.

Networks (SNNs) are employed that encode information more efficiently than Artificial Neural Networks (ANNs), as evidenced by recent research (Singh et al., 2023). Consequently, finding alternatives to gradient-based training methods is paramount for the application of more suitable objectives and more information-efficient models.

## 1.1 Approach

We introduce Layer-wise Feedback Propagation (LFP), a novel principle of training based on eXplainable artificial intelligence (XAI). As summarized in Figure 1, optimizing a model with LFP consists of three distinct steps:

*A: Forward Pass.* In each iteration, LFP first requires the model to perform a forward pass, similar to other training methods such as gradient descent. Here, some intermediate variables such as activations are set and a prediction is obtained.

*B: Feedback Propagation.* Based on this prediction, the model then receives an *initial reward* (cf. Section 3.2) evaluating the prediction quality. In a modified backward pass, LFP decomposes this initial reward into *local rewards* that provide feedback to single neurons and connections.

*C: Weight Update.* Finally, LFP updates parameters based on a local reward at each neuron. In this work, the *local* reward is obtained from step B; however, a local reward obtained from any source could be used in theory. Here, positive and negative reinforcement is leveraged in a greedy manner, strengthening connections that work well (i.e., receive a positive reward) and weakening those that do not (i.e., receive a negative reward). This step is similar to a Hebbian update rule, as discussed in Section 3.1.2.

As such, LFP is related to the reinforcement learning paradigm of encouraging rewarded behavior, implementing it at the smallest elements of neural networks — neurons and connections. Refer to Section 3 for more details.

To propagate feedback to each neuron, we decompose an initial reward (cf. Figure 1) based on each neuron's contribution. The question of neuron contributions (or feature importance) is addressed by a subset of interpretability methods (Baehrens et al., 2010; Simonyan et al., 2013; Springenberg et al., 2015; Bach et al., 2015; Sundararajan et al., 2017)

Figure 2: Venn diagram of the capabilities of LFP and supervised gradient descent. LFP shares some properties of gradient descent but provides several additional benefits, as it does not require gradient computation and thus provides increased flexibility w.r.t. choice of model and objective, and naturally encourages sparsity.

that focus on obtaining *local* explanations. These techniques assign importance scores to individual features or groups of features in specific samples, based on the model's behavior. It is important to note that some of these methods are not limited to attributing importance solely to input features but can also determine the significance of individual units within a model. Among these, LRP (Bach et al., 2015) efficiently explains predictions after inference by computing the contribution of network components to the prediction outcome. This is achieved by decomposing an initial relevance value assigned to the output layer through a modified backward pass. LFP leverages the LRP-decomposition rules to distribute an initial reward during step B in Figure 1 (in place of an initial relevance) to individual neurons during training. Our method then utilizes the distributed reward at each neuron to update parameters (see step C of Figure 1).

Compared to several other gradient alternatives such as evolutionary approaches or methods from the domain of biologically plausible learning, LFP makes more assumptions about the model and objective, for instance, that the model is decomposable in terms of layer-wise mappings or that forward weights equal backward weights. However, our method preserves similar computational complexity, performance and convergence speed to gradient descent (cf. Section 3.3) and scales well to complex tasks and models (cf. Sections 3.5 A.9) but does not require gradient computation (see Figure 2). This allows for increased flexibility in terms of model and objective function compared to gradient-based methods. Consequently, LFP can be employed for training both Deep Neural Networks (DNNs) (cf. Sections 3.5, 3.6) and neuromorphic architectures such as SNNs (cf. Section 4.2). The formulation of objective functions for LFP is similarly flexible. For instance, similar to supervised gradient descent, prior knowledge about the task such as ground truth predictions can be included in the objective. But if such knowledge is not available, or non-differentiable elements should be integrated, a valid objective for LFP can still be formulated (see Table 1). It is therefore well-suited to supervised tasks (and naturally extends to reinforcement learning tasks as well, although we leave this application to future work).

Neurons that do not contribute in the forward pass are not altered by LFP, which encourages sparse (and thus theoretically more efficient) updates, a property also found in biological networks (Pozzi et al., 2020). Due to an implicit weight scaling, the models obtained via LFP sparsely represent information and are easily prunable, contributing to energy-efficient Artificial Intelligence (AI).

## 1.2 Contributions

The primary objective of this work is to introduce LFP and demonstrate that this method provides a valid training signal and converges to a local optimum in a set of controlled tasks explored in this work. Specifically, we focus on the well-understood supervised problem of classification, using fully-connected and convolutional neural networks (although we provide simple initial examples for regression and transformers in Appendix A.8 and A.9, respectively). We further examine the properties of LFP and consequently explore two applications that impact AI energy efficiency and expressivity. We can show that LFP performs well in these semi-complex settings, which is a positive indicator for

Table 1: Overview of model and objective configurations solvable by LFP and gradient-based methods (in the table, *D* stands for Differentiable and *ND* for Nondifferentiable), respectively, in the context of supervised learning. LFP is more flexible and can be applied to a wide variety of learning settings without requiring significant adaptations or substitutions.

| Configuration | | | Example | Gradient-based Training | LFP-based Training |
|---|---|---|---|---|---|
| | *D* | *ND* | | | |
| Objective | ☒ | ☐ | MNIST Classification with | ✓ | ✓ |
| Model | ☒ | ☐ | ReLU-activated MLP | | |
| | *D* | *ND* | | | |
| Objective | ☒ | ☐ | Learning with Neuromorphic | ✗[0] | ✓ |
| Model | ☐ | ☒ | Architecture, e.g. SNNs | | |
| | *D* | *ND* | | | |
| Objective | ☐ | ☒ | Non-differentiable evaluation measures | ✗[1] | ✓ |
| Model | ☒ | ☐ | in objective, e.g. Levenshtein distance | | |
| | *D* | *ND* | | | |
| Objective | ☐ | ☒ | Any Combination of the | ✗[0] | ✓ |
| Model | ☐ | ☒ | Two Rows Above | | |

a successful extension to more difficult problems as well, given appropriate parameterization and further research. The current work thus serves as a foundation for understanding the fundamental aspects of LFP, with future investigations expected to explore the method's applicability in more state-of-the-art scenarios.

In summary, we make the following contributions:

1. We propose Layer-wise Feedback Propagation (LFP), a novel, interpretability-based training paradigm for DNNs that updates parameters based on a *local*, neuron-wise reward.

2. We prove the convergence of LFP and confirm this theoretical result empirically on several different models and datasets.

3. We discuss the properties, advantages and disadvantages of LFP.

4. Based on the discovered properties, we investigate two applications of LFP in-depth: Obtaining sparser models, that represent information efficiently and can be pruned more easily, and training non-differentiable models, such as Heaviside-activated SNNs.

## 2 Related Work

### 2.1 Deep Learning Approaches

Depending on the amount of (human) supervision involved, deep learning approaches can generally be categorized into supervised, unsupervised, and reinforcement learning, as well as hybrids between these, such as semi-supervised approaches. Setting aside unsupervised and hybrid methods, supervised approaches assume the availability of examples with ground truth predictions and aim to solve a well-defined and specific task. Reinforcement learning makes far less assumptions, requiring a model to learn to interact with a potentially changing environment only from a reward signal that is typically sparse and discrete. As such, reinforcement learning is the most general category, as most supervised problems can be reformulated as reinforcement learning problems, while the reverse is usually not possible. However, this reformulation is often inefficient, since it leverages less domain knowledge and results in an optimization problem

---

[0]Gradient-based methods are not applicable here without substituting non-differentiable model parts in the backward pass

[1]These settings can be expressed as reinforcement learning problems and solved through gradient-based reinforcement learning methods — however, depending on the specific objective, this reformulation can increase problem difficulty and thus lead to suboptimal solutions (Barto & Dietterich, 2004).

that is more difficult to solve (Barto & Dietterich, 2004). Therefore, if the task constraints allow, supervised methods are preferred in practice.

Within these categories, gradient-based approaches (Rumelhart et al., 1985; Sutton et al., 1999; Mnih et al., 2013; Schulman et al., 2017) are by far the most widely used. However, as detailed in Section 1, these approaches assume the model and (for supervised learning) the objective function to be differentiable w.r.t. the model parameters. This is not always desirable because, as a consequence, gradient-based learning restricts the space of usable models, can be inefficient (Zuo, 2020), limits implementations on hardware, and is not plausible from a biological perspective (Pozzi et al., 2020; Hinton, 2022). Several alternative approaches have been proposed to address these issues over the years. For instance, a number of studies (Goldstein et al., 2014; Carreira-Perpiñán & Wang, 2014; Taylor et al., 2016; Zeng et al., 2019; Wang et al., 2019; Tang et al., 2020; Wang et al., 2022) formulate neural network training as alternating minimization problems using auxiliary variables. These methods make training easily parallelizable and require comparatively few (but computationally expensive) iterations, but often scale badly to complex problems. Another set of methods (Salimans et al., 2017; Such et al., 2017; Cui et al., 2019; Tripathi & Singh, 2020; Chen et al., 2021) involves training models through evolutionary strategies, by making random alterations to the model's parameters and selecting the best-performing version. While these strategies are highly flexible w.r.t. model architecture and task description, they are based on sampling strategies and therefore do not scale well to larger models.

Several learning methods specifically aim for higher biological plausibility. For instance, feedback alignment (Lillicrap et al., 2016; Nøkland, 2016) removes the need for symmetric weights in the forward and backward pass (a biologically implausible property), but otherwise backpropagates gradients nonetheless, limiting its applicability to e.g. non-differentiable models. Variations of target propagation (Le Cun, 1986; Lee et al., 2015; Bartunov et al., 2018; Ernoult et al., 2022; Roulet & Harchaoui, 2023) introduce dedicated backward connections to directly generate layer-wise activation targets and update parameters locally per layer based on these targets. This avoids the need for both symmetric weights and error propagation altogether and can even pass meaningful training signals through non-differentiable activations (Lee et al., 2015), but requires optimizing an additional, dedicated feedback path. Approaches to Hebbian learning (Hebb, 1949; Whittington & Bogacz, 2017; Miconi, 2021; Gupta et al., 2021; Lagani et al., 2022; Demidovskij et al., 2023; Journé et al., 2023) implement fully local but unsupervised learning rules proportional to the product of neuron inputs and outputs. To extend Hebbian learning to supervised settings, it is usually combined with backpropagation of a feedback signal Gupta et al. (2021); Demidovskij et al. (2023) in order to obtain both local and global representations. Finally, Hinton (2022); Ghader et al. (2025) suggest a contrastive approach to learning that requires no backward pass but assumes data to be split in positive and negative examples.

For specialized architectures, gradient application can be especially inefficient or infeasible, resulting in the development of dedicated learning methods for these settings. In the context of convergent recurrent architectures and Hopfield-like models, approaches such as contrastive Hebbian learning (Movellan, 1991; Detorakis et al., 2019) and equilibrium propagation (Scellier & Bengio, 2017; Ernoult et al., 2019; Laborieux et al., 2021) approximate local gradients through differences in neural activities between a (free) forward phase and a backward phase where the network outputs are nudged towards a target. Due to the need for two settling phases, these approaches are comparatively slow but effectively avoid the computation of gradients, inspiring extensions to physical systems such as SNNs (Martin et al., 2021) or Ising Machines (Laydevant et al., 2024), where gradient-based learning is notoriously difficult. Due to their promising properties Singh et al. (2023), specifically gradient-free optimization on SNNs is an active field of research. Here, DECOLLE (Kaiser et al., 2020) avoids propagation through time by minimizing the global loss at each layer separately using random readout matrices, and then utilizes surrogates to approximate Heaviside gradients. The unsupervised spike-time dependent plasticity (Bengio et al., 2015; Mozafari et al., 2018) and its extensions for supervised settings, e.g., Reward-modulated Spike-Timing Dependent Plasticity (R-STDP) (Frémaux & Gerstner, 2016), implement a local, unsupervised learning rule akin to Hebbian principles that considers the timing between incoming and outgoing spikes.

Our method is biologically implausible, as it requires symmetric forward and backward weights and backpropagates a reward signal. However, it implements select properties from biological learning, such as only altering contributing neurons Pozzi et al. (2020) and performing local, hebbian-like updates modulated by a reward (cf. Section 3.1.2). LFP scales well to large-scale tasks (cf. Sections 3.5, 4.1) and avoids gradient computations altogether by adopting a multi-agent view of DNNs and distributing a global reward locally to single neurons based on their contribution. It is therefore closely related to techniques that apply reinforcement to single neurons, either based on a global reward (Seung, 2003) or by assuming local, neuron-wise rewards are available (Wang & Cai, 2015). Among these methods, BrainProp (Pozzi et al., 2020) also distributes a feedback signal to single neurons based on a global reward, similar

to LFP. However, BrainProp's feedback consists of an error signal derived from the global reward, and consequently requires computation of derivatives and differentiable models. Furthermore, the method optimizes for one output unit at a time, limiting convergence speed, and the feedback can grow arbitrarily large for deeper models due to long chains of weight multiplications. In contrast, LFP does not require models to be differentiable as no gradients are computed, implicitly normalizes the distributed reward (cf. Equation 2), and can optimize for all outputs at the same time. This makes LFP applicable to both supervised and reinforcement learning problem formulations.

## 2.2 Utility of Intermediate Explanations

Due to the highly nonlinear complexity of DNNs, their underlying decision-making is difficult to interpret. For this reason, the field of XAI has come forth with several methods to elucidate on a model's behavior. Here, *post-hoc* methods aim to explain existing and already trained models and can be divided into *global* and *local* approaches:

*Global* XAI interprets a model's behavior in a general manner, e.g., by identifying (and visualizing) encoded concepts or learned representations and their hierarchical relationships (Bau et al., 2017; Kim et al., 2018; Hu et al., 2020; Hohman et al., 2020; Zhang et al., 2021; Fel et al., 2023b; Achtibat et al., 2023; Vielhaben et al., 2023; Kowal et al., 2024), by computing (and visualizing) its sensitivities (Santurkar et al., 2019; Fel et al., 2023a), or by interpreting their internal mechanisms through surrogate models (Huben et al., 2024; Gorton, 2024; Bhalla et al., 2024; Dreyer et al., 2025).

*Local* XAI instead explains the individual predictions of a model. That is, for a particular sample, local explanation techniques seek to assign importance scores to individual features or groups of features that have the most influence on a model's decision. Here, sampling-based techniques (Štrumbelj & Kononenko, 2014; Ribeiro et al., 2016; Zintgraf et al., 2017; Fong & Vedaldi, 2017; Lundberg & Lee, 2017) assume models to be impenetrable black boxes, and consequently explain through a challenge-response-scheme involving perturbation, but are costly to compute and approximative. Methods that modify the backward pass, such as Baehrens et al. (2010); Bach et al. (2015); Montavon et al. (2017); Shrikumar et al. (2017); Sundararajan et al. (2017) require grey-box access to a model's internal parameters. In turn, most of these latter methods can be implemented to explain efficiently in terms of computation time (requiring exactly one forward-backward pass).

Beyond the human-understandable explanations in input space, a secondary property of *local* XAI is the ability to generate intermediate explanations, i.e., assign importance scores to the activations (or for some methods even to weighted connections) of intermediate layers. Especially methods that modify the backward pass yield such intermediate explanations naturally and efficiently, as a by-product of computing input-level scores.

Several previous works have utilized these intermediate importance assignments towards improving models: For instance, explanation scores can be employed similarly to an attention mechanism during training, masking features in the forward pass (Fukui et al., 2019; Schiller et al., 2019; Sun et al., 2020), or to guide which information is passed through dropout layers beyond random choice (Zunino et al., 2021). Lee et al. (2019); Nagisetty et al. (2020) instead apply a similar XAI-based masking to the gradients during the backward pass. In trained models, intermediate explanations are employed as a criterion for pruning or quantization (Voita et al., 2019; Sabih et al., 2020; Becking et al., 2022; Yeom et al., 2021; Soroush et al., 2023; Hatefi et al., 2024) to increase efficiency. Similarly, Ede et al. (2022) proposes XAI-guided plasticity regularization of relevant neurons in order to mitigate catastrophic forgetting when learning consecutive tasks.

Our method is related to the above works that leverage intermediate explanations for model improvement both during and after training, specifically those that augment the backward pass (Lee et al., 2019; Nagisetty et al., 2020). However, in contrast to the above methods, we do not utilize explanations as additional guidance, and rather distribute a reward directly to single neurons based on their contribution. Instead of explaining a specific decision, this can be understood as explaining the source of an obtained reward in terms of internal model components and updating parameters based on that information. We choose to utilize Layer-wise Relevance Propagation (LRP) for reward propagation due to its demonstrated utility in several of the above applications (Lee et al., 2019; Sun et al., 2020; Becking et al., 2022; Yeom et al., 2021; Ede et al., 2022; Soroush et al., 2023; Hatefi et al., 2024).

## 3 Layer-wise Feedback Propagation

In this section, we introduce the methodology behind LFP in detail, demonstrate its convergence empirically, and prove it theoretically.

### 3.1 LFP Backpropagation and Credit Assignment

LFP assigns credit to parameters by distributing a reward throughout the model. We detail the specific LFP backpropagation rules — partially inherited from the LRP methodology — below:

#### 3.1.1 Reward Propagation and Parameter Update

Let $x$ be a singular input sample to model $f$, with corresponding one-hot encoded ground-truth label $y \in \{0, 1\}^C$, and $C$ possible classes. Then, $f(x)$ denotes the model output given $x$ and $a_j^{[l+1]}$ the output of an arbitrary neuron $j$ at layer $l + 1$. This output is computed as

$$a_j^{[l+1]} = \phi_j^{[l+1]} \left( \sum_i w_{ij}^{[l+1]} \cdot a_i^{[l]} \right) , \tag{1}$$

where $\phi_j^{[l+1]}$ denotes the activation function of the neuron $j$ and the parameters $w_{ij}^{[l+1]}$ weigh the output $a_i^{[l]}$ of the $i$-th neuron at layer $l$. The weight $w_{ij[l+1]}$ may be 0 if there is no connection between $i$ and $j$. We view the bias $b_j^{[l+1]}$ of each neuron $j$ as a weighted connection constantly receiving $a_{b_j}^{[l+1]} = 1$ in this context. As common in LRP-literature (e.g., Montavon et al. (2019)), we denote the contribution of a connection between neurons $i, j$ as $z_{ij}^{[l+1]} = w_{ij}^{[l+1]} \cdot a_i^{[l]}$ and the pre-activation of the $j$-th neuron as $z_j^{[l+1]} = \sum_i z_{ij}^{[l+1]}$.

LFP then assumes that an *initial reward* $r_c$ is available for each output neuron $c$, measuring how well it contributes to solving the task. In practice, $r_c$ is decomposed from a *global reward* $R_c$ assigned to the predicted outcome (e.g., the class in supervised classification) corresponding to $c$. This initial decomposition from $R_c$ to $r_c$ may be task-specific. Refer to Section 3.2 for more details. Intuitively, a positive reward should cause positive reinforcement, and a negative reward negative reinforcement. Therefore, the resulting change in neuron activation depends on the sign of both the reward and the activation. To optimize performance, if a positively activated neuron receives positive or negative reward, we signal it to increase or decrease, respectively. With the same-signed reward, a negative activation is signaled to decrease or increase, respectively. After obtaining $r_c$, it is propagated backwards through the model based on the backpropagation rules of LRP, so that the at layer $l + 1$ eventually receives a *local reward* $r_j^{[l+1]}$, as a decomposition of the initial $r_c$ of all $c \in C$. In the below derivations, the LRP-0-rule is used for simplicity, although this rule is generally not applicable without issue in practice due to numerical instability (cf. Sections below). Then, a reward message $r_{ij}^{[l+1]}$ is sent from neuron $j$ at layer $l + 1$ to neuron $i$ at layer $l$ (i.e., through the connection weighted by $w_{ij}^{[l+1]}$):

$$r_{ij}^{[l+1]} = \frac{z_{ij}^{[l+1]}}{z_j^{[l+1]}} \cdot r_j^{[l+1]} , \tag{2}$$

The *local* reward for neuron $i$ then accumulates as

$$r_i^{[l]} = \sum_j r_{ij}^{[l+1]} . \tag{3}$$

Equation 3 inherently assigns credit to $i$, providing a notion of how helpful an activated neuron or a parameter is in solving the task correctly. Similarly, $r_{ij}^{[l+1]}$ from Equation 2 is derived by decomposition from the local reward $r_c$, and can thus be interpreted as a reward for the connection weighted by $w_{ij[l+1]}$. Note that (excluding reward flowing to bias terms) Equation 3 and Equation 2 implicitly normalize the decomposed reward to sum up to the initial reward (cf. Bach et al. (2015)). I.e., if the model has $L$ layers $l \in [0, 1, ..., L - 1]$, then $\forall l. \sum_i r_i^{[l]} = \sum_j r_j^{[l+1]} = \sum_c r_c^{[L]}$ holds.

In order to derive an update for $w_{ij}^{[l+1]}$, we replace $w_{ij}^{[l+1]}$ in $z_{ij}^{[l+1]}$ from Equation equation 2 with $\text{sign}(w_{ij}^{[l+1]}) \cdot w_{ij}^{[l+1]} = |w_{ij}^{[l+1]}|$. I.e., we multiply the reward $r_{ij}^{[l+1]}$ with the parameter sign (since identically signed rewards should result in opposite update directions for negative/positive parameters) and thus arrive at the following update step:

$$d_{w_{ij}^{[l+1]}}^{\text{lfp}} = \frac{|w_{ij}^{[l+1]}| \cdot a_i^{[l]}}{z_j^{[l+1]}} \cdot r_j^{[l+1]}$$

$$(w_{ij}^{[l+1]})^{\text{new}} = w_{ij}^{[l+1]} + \eta \cdot d_{w_{ij}^{[l+1]}}^{\text{lfp}} \tag{4}$$

Here, $\eta$ is a learning rate. We refer to this basic formulation of LFP as LFP-0. Note that while we utilize LRP-like rules to obtain $r_j^{[l+1]}$, this is not strictly required, and $r_j^{[l+1]}$ could theoretically come from any source, with Equation 4 performing a reward-modulated Hebbian-like update, as detailed below. Optimization akin to Stochastic Gradient Descent (SGD) with a momentum can then be described as follows:

$$d_{w_{ij}^{[l+1]}}^{\text{mom}} = d_{w_{ij}^{[l+1]}}^{\text{old}}$$

$$d_{w_{ij}^{[l+1]}} = \alpha \cdot d_{w_{ij}^{[l+1]}}^{\text{mom}} + (1 - \alpha) \cdot d_{w_{ij}^{[l+1]}}^{\text{lfp}}$$

$$(w_{ij}^{[l+1]})^{\text{new}} = w_{ij}^{[l+1]} + \eta \cdot d_{w_{ij}^{[l+1]}} \tag{5}$$

where $\alpha \in [0, 1]$.

Since compared to SGD, LFP employs a reward instead of a loss function, it requires maximization instead of minimization in the update step of the above formula. This reward is then distributed throughout the model, which — as detailed by the above equations — requires no gradient computation[2]. Consider e.g. the Heaviside step function (cf. Figure A.11): This activation function has zero or infinite gradients everywhere, but Equation 2 and Equation 3 can still propagate a meaningful non-zero signal through models with Heaviside activations.

### 3.1.2 Connection to Hebbian Learning

While we derived equations (2–4) from the relevance propagation of LRP, as described in Bach et al. (2015), there is a close similarity of the resulting update rule to Hebbian learning (Hebb, 1949). At its core, Hebbian learning formulates weight updates as

$$d_{w_{ij}^{[l+1]}}^{\text{Hebb}} = a_i^{[l]} a_j^{[l+1]}. \tag{6}$$

The LFP update rule from Equation 4 can be reformulated as follows:

$$d_{w_{ij}^{[l+1]}}^{\text{lfp}} = \frac{|w_{ij}^{[l+1]}| \cdot a_i^{[l]}}{z_j^{[l+1]}} \cdot r_j^{[l+1]}$$

$$\propto \frac{a_i^{[l]}}{z_j^{[l+1]}} \cdot r_j^{[l+1]} \ |a_j^{[l+1]} \approx z_j^{[l+1]} \tag{7}$$

$$\approx \frac{a_i^{[l]}}{a_j^{[l+1]}} \cdot r_j^{[l+1]}$$

Since (excluding 0) $\text{sign}(a_i^{[l]}/a_j^{[l+1]}) = \text{sign}(a_i^{[l]} \cdot a_j^{[l+1]})$, the LFP weight update can be interpreted as a Hebbian update, modulated by the local reward $r_j^{[l+1]}$. In expectation, Equation 6 simply results in increasing $|a_j^{[l+1]}|$. By multiplying with $r_j^{[l+1]}$, LFP contextualizes $a_j^{[l+1]}$ to a given task, decreasing $|a_j^{[l+1]}|$ if neuron $j$ hinders performance as evaluated

---

[2]Equation 2 can be written as activation times gradient only for (piece-wise) linear models, e.g., with ReLU activations, cf. Theorem 1

by the reward function. Aside from this reward modulation, the largest difference to the original Hebbian formulation is the division of $a_i^{[l]}$ and $a_j^{[l+1]}$ in place of multiplication. While this preserves the update direction, it also implicitly regularizes the update when the weights grow too large (and, as a consequence, also $|a_j^{[l+1]}|$), similar to, e.g., Oja's rule (Oja, 1989). Lastly, the LFP-update includes a multiplication with the weight magnitude that induces sparse updates, which have been shown to improve performance in the context of Hebbian learning (Gupta et al., 2021).

### 3.1.3  Special Case: Numerical Stability

The LFP-0-rule described above captures the fundamental idea of how rewards are distributed via LFP but is infeasible in practical applications due to being numerically unstable for denominators close to zero (cf. the LRP-0-rule (Bach et al., 2015)).

Inspired by Bach et al. (2015), we therefore employ a normalizing constant $\varepsilon$ to increase numerical stability, resulting in the *LFP-ε-rule*:

$$r_{ij}^{[l+1]} = \frac{z_{ij}^{[l+1]}}{z_j^{[l+1]} + \text{sign}(z_j^{[l+1]}) \cdot \varepsilon} \cdot r_j^{[l+1]} \tag{8}$$

$$d_{w_{ij}^{[l+1]}}^{\text{lfp}} = \frac{|w_{ij}^{[l+1]}| \cdot a_i^{[l]}}{z_j^{[l+1]} + \text{sign}(z_j^{[l+1]}) \cdot \varepsilon} \cdot r_j^{[l+1]} \tag{9}$$

Note that for the remainder of this work, we define $\text{sign}(0) = 1$. We utilize this rule in all experiments as a replacement for LFP-0. As recommended by Bach et al. (2015), we choose $\varepsilon = 10^{-6}$.

### 3.1.4  Special Case: Batch Normalization Layers

Attribution methods based on modified backpropagation, such as LRP, are generally not implementation invariant and require models to be canonized (i.e. restructuring models to a functionally equivalent version that only contains layer types with well-understood properties w.r.t. a given explanation method) before the explanations are computed (Kohlbrenner et al., 2020; Motzkus et al., 2022), which usually involves merging multiple layers. This mostly affects BatchNorm layers (Ioffe & Szegedy, 2015) (and their neighboring layers), which implement the following operation:

$$(a_j^{[l+1]})^{bn} = \gamma_i^{[l+1]} \frac{a_i^{[l]} - \mathbb{E}[a_i^{[l]}]}{\sqrt{\text{Var}[a_i^{[l]}] + \epsilon}} + \beta_i^{[l+1]}$$

We expect the above issue to extend to LFP as well due to its close relation to LRP. However, merging BatchNorm layers during training is not possible, as these layers also contain learnable parameters. For this reason, we instead provide a rule to propagate reward through BatchNorm layers approximatively by first writing the BatchNorm equation as two consecutive linear operations:

$$(a_j^{[l+1]})^{bn} = \gamma_i^{[l+1]} \tilde{a}_i^{[l]} + \beta_i^{[l+1]} \text{ and } \tilde{a}_i^{[l]} = \frac{1}{\sqrt{\text{Var}[a_i^{[l]}] + \epsilon}} a_i^{[l]} - \frac{\mathbb{E}[a_i^{[l]}]}{\sqrt{\text{Var}[a_i^{[l]}] + \epsilon}} \tag{10}$$

This allows us to formulate the following LFP rule:

$$r_{ij}^{[l+1]} = \frac{\gamma_i \cdot \frac{a_i^{[l]}}{\sqrt{\text{Var}[a_i^{[l]}] + \epsilon}}}{a_j^{[l+1]} + \text{sign}(a_j^{[l+1]}) \cdot \varepsilon} \cdot r_j^{[l+1]} \tag{11}$$

Table 2: Examples of reward functions for multiclass classification (i.e., one output neuron per class) with LFP. In the table, $\sigma$ denotes the sigmoid function, and SM the softmax function. *GD* denotes gradient descent and *CE* cross-entropy.

| ID | $r_c$ | $\rho(f(x))$ | $R_c$ | Theorem 1: *GD*-Equivalent |
|---|---|---|---|---|
| #1 | $\text{sign}(o_c) \cdot \mathbb{1}(y_c = 1)$ | $1$ | $\mathbb{1}(y_c = 1)$ | ✗ |
| #2 | $\text{sign}(o_c) \cdot$ $-\mathbb{1}(\exists c'.y_{c'} = 1 \wedge o_c > o_{c'})$ | $1$ | $-\mathbb{1}(\exists c'.y_{c'} = 1 \wedge o_c > o_{c'})$ | ✗ |
| #3 | $o_c \cdot (y_c - \text{SM}(o_c))$ | $\begin{cases} (1 - \text{SM}(o_c)) & \text{if } y_c = 1 \\ \text{SM}(o_c) & \text{if } y_c = 0 \end{cases}$ | $\begin{cases} \lvert o_c \rvert & \text{if } y_c = 1 \\ -\lvert o_c \rvert & \text{if } y_c = 0 \end{cases}$ | Softmax-*CE* |
| #4 | $o_c \cdot (y_c - \sigma(o_c))$ | $\begin{cases} (1 - \sigma(o_c)) & \text{if } y_c = 1 \\ \sigma(o_c) & \text{if } y_c = 0 \end{cases}$ | $\begin{cases} \lvert o_c \rvert & \text{if } y_c = 1 \\ -\lvert o_c \rvert & \text{if } y_c = 0 \end{cases}$ | Sigmoid-*CE* |

$$d^{\text{lfp}}_{\gamma_i^{[l+1]}} = \frac{\lvert \gamma_i^{[l+1]} \rvert \cdot \tilde{a}_i^{[l]}}{a_j^{[l+1]} + \text{sign}(a_j^{[l+1]}) \cdot \varepsilon} \cdot r_j^{[l+1]}$$

$$d^{\text{lfp}}_{\beta_i^{[l+1]}} = \frac{\lvert \beta_i^{[l+1]} \rvert}{a_j^{[l+1]} + \text{sign}(a_j^{[l+1]}) \cdot \varepsilon} \cdot r_j^{[l+1]} \tag{12}$$

For updating parameters $\gamma_i^{[l+1]}$ and $\beta_i^{[l+1]}$, the above rule simply applies LFP-$\varepsilon$ to the second, parameterized linear operation. This is also done for passing the reward backward in Equation 11, but a mean $\mathbb{E}(a_i^{[l]}) = 0$ is assumed for this backward pass since shifting rewards across 0 can lead to updates in the wrong direction in preceding layers. Note that $z_j^{[l+1]} = a_j^{[l+1]}$ here, since BatchNorm employs a linear activation function.

We apply this rule to the ResNet (He et al., 2016) and VGG-16-BN (Simonyan & Zisserman, 2014) models in our experiments.

### 3.1.5 Special Case: Skip Connections

Skip connections do not contain parameters and simply merge incoming activation signals from multiple pathways. They are present in several architectures such as ResNets (He et al., 2016) and merge activations either through concatenation $(a_j^{[l+1]})^{\text{skip}} = [a_i^{[l]}, a_k^{[l-l']}]$ or summation $(a_j^{[l+1]})^{\text{skip}} = a_i^{[l]} + a_k^{[l-l']}$. For the former type, we pass rewards through the concatenation. For the latter type, rewards are decomposed according to their proportional contributions:

$$r_{ij}^{[l+1]} = \frac{a_i^{[l]}}{a_j^{[l+1]} + \text{sign}(a_j^{[l+1]}) \cdot \varepsilon} \cdot r_j^{[l+1]}$$

$$r_{kj}^{[l+1]} = \frac{a_k^{[l-l']}}{a_j^{[l+1]} + \text{sign}(a_j^{[l+1]}) \cdot \varepsilon} \cdot r_j^{[l+1]} \tag{13}$$

## 3.2 Obtaining Initial Rewards

LFP distributes a reward signal $r_c$ throughout the model. $r_c$ is the *initial reward* assigned to the output neuron $c$. As such, it slightly differs from a *global reward* $R_c$ (as employed, e.g., in reinforcement learning) assigned to the outcome associated with neuron $c$. We distinguish between an *output* and an *outcome* as follows: For instance, consider the example of (multiclass) classification. The highest-activating output neuron $c = 0$ may produce the output $o_c = 1.234$[3]

---

[3]Note that we assume output neurons to be linearly activated (i.e. $o_c = z_c$) in all models except SNNs, where we assume a Heaviside activation. For multiclass classification, the class with the largest $o_c$ is predicted, and any nonlinearities (e.g., softmax to obtain probabilities) are interpreted as part of the objective function.

for a given sample. Consequently, the class corresponding to this neuron is chosen as a prediction, resulting in the *outcome* that class $c = 0$ is assigned to the given input. I.e., the outcome (the class prediction in this example) is derived from the output (the raw logit) via some rule or function that depends on the task. Note that the current prediction in the above example is only one of $C$ possible outcomes, and more than one outcome may be considered at the same time by obtaining nonzero $R_c$ for more than one class.

In this framework, $R_c$ measures the quality of an outcome, not considering the model that produced it. For instance, the correctness of a class assignment to a given input. $r_c$ considers the model by decomposing $R_c$ to the output $o_c$ of the neuron $c$, based on the mapping from output to outcome. Consequently, how $o_c$ should be evaluated w.r.t. the task depends on how an outcome (e.g., a class prediction) is derived from $o_c$. For instance, consider binary classification with an instance correctly classified as class 0, and $R_0 = 1$. If two output neurons exist, each corresponding to a single class, neuron $c = 0$ receives a positive global reward. However, if $o_0 < 0$, the initial reward $r_0$ should become negative in order to encourage a positive output maximizing class probability. On the other hand, if only one output neuron exists and the sign of $o_0$ determines the predicted class, $r_0$ should be positive, e.g., $r_0 = R_0$, in order to encourage the current (correct) direction of $o_0$ — again maximizing class probability. As shown in the example, obtaining $r_c$ from $R_c$ therefore requires an initial decomposition that depends on the relationship between output $o_c$ and the associated outcome.

For supervised classification, which is the focus of this work, deriving $r_c$ from $R_c$ is straightforward: In the special case of binary classification with one output neuron, we maintain $r_c = R_c$. Otherwise, for multiclass settings with one output neuron per class, $r_c$ can be obtained as

$$r_c = \text{sign}(o_c) \cdot R_c \tag{14}$$

Since the reward directly affects parameter updates (cf. Equation 4), the model will be updated as long as long as a $r_c$ is nonzero. While this is not the only solution[4], including a convergence mechanism $\rho(f(x)) : \mathbb{R}^C \mapsto [0, 1]$ in the initial reward $r_c$ alleviates this issue:

$$r_c = \text{sign}(o_c) \cdot \rho(f(x)) \cdot R_c \tag{15}$$

Under the above formulation, possible reward functions for LFP are listed in Table 2. In this table, the third and fourth row are equivalent to commonly used loss functions for gradient descent, as described in Theorem 1 (cf. Section 3.3). Specifically, #3 corresponds to a cross-entropy loss with softmax, and #4 to a cross-entropy loss with sigmoid. Note that for the purpose of Theorem 1, we view the softmax and sigmoid nonlinearities as part of the loss function, applied to linearly activated outputs $o_c$. We employ reward #3 in our experiments. We briefly investigate LFP with reward #2 in Appendix A.3.

## 3.3 Convergence of LFP

In this section, we demonstrate theoretically and experimentally in the context of supervised classification that LFP converges, and can successfully train ML models. First, we show the following theorem:

**Theorem 1.** *For a differentiable loss function L and any ReLU-activated network, LFP-0 with initial reward*

$$r_c = o_c \cdot \left( -\frac{d\mathcal{L}}{do_c} \right)$$

*is equivalent to weight-scaled gradient descent, i.e.*

$$(w_{ij}^{[l+1]})^{new} = w_{ij}^{[l+1]} - \eta \cdot |w_{ij}^{[l+1]}| \cdot \frac{d\mathcal{L}}{dw_{ij}^{[l+1]}}.$$

---

[4]There exist heuristics for training despite non-converging rewards, such as zero-capped rewards, capped parameters, early stopping, or progressively decaying learning rates or rewards. Utilization of such techniques allows for the application of a wider range of reward functions beyond those applied in this work.

*In particular, allowing for individual step sizes $\eta_{ij}$ if we choose $\eta_{ij} = |w_{ij}^{[l+1]}|^{-1} \cdot \eta$ we can frame gradient descent for any differentiable loss function in the LFP-framework.*

For the detailed proof, refer to Appendix A.2. While we proved the statement in the Theorem for ReLU-activated models, as that is by far the most common setup in DNNs, it in fact holds for several piecewise linear activation functions (such as Linear, ReLU, or Leaky ReLU). I.e., for networks with these activation functions, we can reformulate gradient descent with any differentiable loss function as LFP-0 with the appropriate reward.

However, LFP can also be used to achieve learning behaviour different from gradient descent as the above theorem is subject to several conditions, which are not necessarily fulfilled in practice (including the experiments below). These conditions are as follows:

**LFP-0 Rule.** The theorem assumes LFP-0 Rule. In practice, we employ LFP-$\varepsilon$ in all experiments for numerical stability. This results in updates that do not only differ in magnitude from LFP-0, but potentially even in sign due to cascading effects from higher layers. Additional rules may be formulated for LFP, which can yield diverse learning dynamics and behaviors (see Appendix A.5).

**Specific Activations.** The theorem implicitly assumes activations

$$\phi_i^{[l]}(z_i^{[l]}) = \frac{d\phi_i^{[l]}(z_i^{[l]})}{dz_i^{[l]}} \cdot z_i^{[l]} \ . \tag{16}$$

This condition is fulfilled by e.g. ReLU, LeakyReLU and Linear activations, but not Heaviside, Tanh, SiLU, ELU, GeLU. For the latter group, LFP may exhibit update behavior different to gradient descent even in simple settings, as shown, e.g. in Appendix A.4 In our experiments, we apply both types of activations.

**Specific Reward Functions.** The theorem only holds for rewards $r_c = o_c \cdot \left(-\frac{d\mathcal{L}}{do_c}\right)$. However, there exist sensible rewards for LFP which cannot be expressed through gradient descent and would require other, reinforcement-learning-based approaches instead, e.g. Table 2, reward #2:

$$r_c = -\mathbb{1}(\exists c'.y_{c'} = 1 \wedge o_c > o_{c'}) \cdot \text{sign}(o_c) \tag{17}$$

where reward -1 punishes any false positive[7].

In a first experiment, we show empirically that LFP is able to train models in simple supervised classification settings. For this purpose, we train a small ReLU-activated MLP (fulfilling the proof-conditions of Theorem 1 except for the LFP-0 rule) on three toy datasets (cf. Figure 3 *left*) using LFP-$\varepsilon$ (Equation 8). For more details about the specific setup, refer to Appendix A.7.1. Figure 3 shows the decision boundaries, accuracies, and weight distributions of the resulting models. Firstly, as shown by the train set accuracies over iterations (*second-to-right*), LFP is able to converge well to respective solutions for the given toy tasks, with the obtained solutions achieving high test set accuracies (*second-to-left*) and their decision boundaries (*left*) separating well between classes. Refer to Appendix A.1 for decision boundaries for other (non-ReLU) activation functions. Secondly, when comparing the performance (*second-to-left*) of LFP and stochastic gradient descent (Grad), we find that they achieve the same accuracy, however, LFP scales differently w.r.t. learning rate (for same performance, $\eta_{\text{LFP}} \approx \frac{\eta_{\text{Grad}}}{100}$). Thirdly, we observe that the solutions (parameters of the model) obtained by LFP are of increased $\theta$-sparsity (*right*), with few large positive or negative parameters when compared to the initialization or solutions obtained by standard gradient descent. When dividing weights into groups via a threshold $\theta^5$, LFP-trained models have much more "neutral" weights w.r.t. $\theta$ than the initial model or the models trained via gradient descent. As shown in Figure 4, this sparsity is caused by the implicit weight-scaling in the LFP update step (cf. Theorem 1). We investigate this property in detail in Section 4.1.

## 3.4 Computational Cost of LFP

LFP leverages LRP to decompose rewards to all neurons in a network. Updates for each connection are then directly derived from the resulting local rewards at each neuron via multiplication with a scalar factor. Consequently, the computational complexity of a single model update with LFP is the same as computing attributions via LRP (and the

---

[7]We investigate LFP with this reward function in Appendix A.3

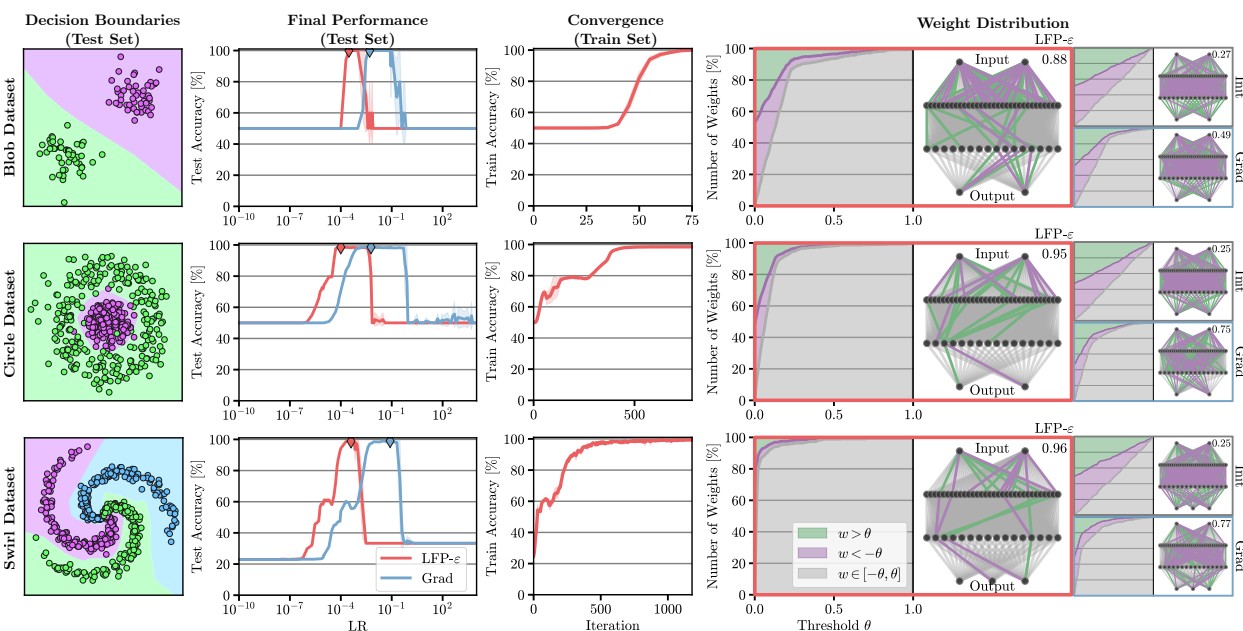

Figure 3: Proof-of-Concept results for LFP on toy data. *Left*: Training with LFP results in the expected decision boundaries. *Second-to-Left*: Under similar training conditions, LFP achieves comparable performance to gradient descent, although it scales differently in terms of learning rate. The diamond marks the highest performance across learning rates; the remaining panels on the right are plotted for this best performing configuration. *Second-to-Right*: LFP converges in a stable manner. *Right*: Distribution of weight values, given a relative threshold $\theta^5$. In each panel, the percentage of large positive (*green*), large negative (*purple*) and neutral (*grey*) weights for each threshold is shown on the left (areas are drawn cumulatively to sum up to one, with lines highlighting the border between areas). On the right of each panel, connections are colored accordingly, for $\theta = 0.25$. Compared to the initialization and the solution found by gradient descent, LFP finds $\theta$-sparse solutions, with more weights being assigned to "neutral" at lower thresholds (number in the top right).

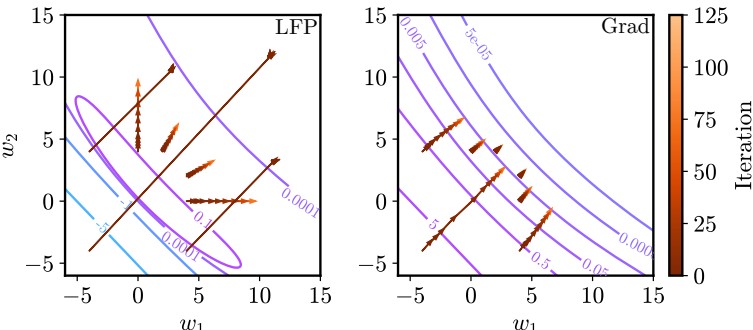

Figure 4: Update behavior of LFP-$\varepsilon$ and gradient descent in a two-parameter toy setting ($w_1$ and $w_2$, no bias). The contours show constant values of the respective objective function (reward *left* and loss *right*, with lighter shades of magenta denoting more positive values and lighter shades of blue denoting more negative values). The training progress of seven differently initialized models is shown in orange. The same initializations were used for both training schemes. Gradient descent simply updates all models towards the global optimum, as measured by the lowest loss. In contrast, LFP distributes a reward throughout the model, updating parameters depending on their contribution. This results in an implicit weight scaling, affects step size and update direction, and encourages finding a sparse solution, where possible. Refer to Appendix A.7.1 for experimental details.

same as computing a gradient descent update), which is in $\mathcal{O}(1)$ given a single forward pass (Achtibat et al., 2024). I.e., theoretically, LFP requires one forward and one backward pass to update the model given a single batch of data. Note

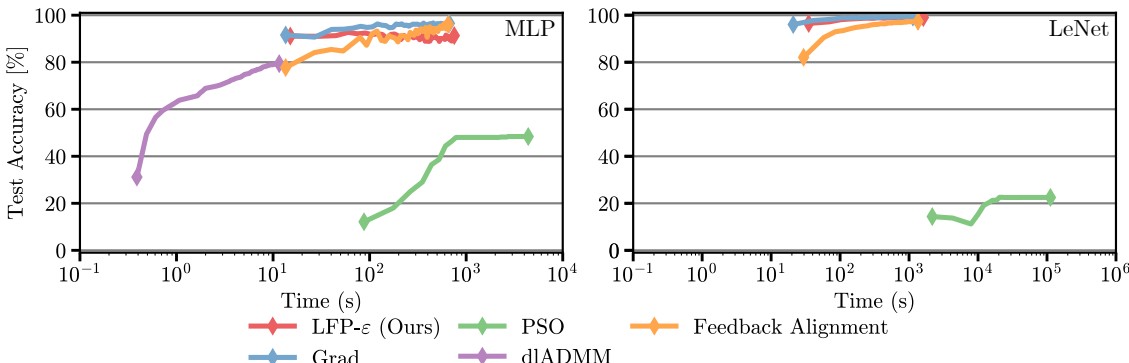

Figure 5: Comparison of test accuracy over time when training an Multi Layer Perceptron (MLP) *(left)* and LeNet *(right)* using LFP vs. various other optimization methods. The training time of LFP is in the same order of magnitude a for other training approaches that perform a backward pass. Note that an implementation of dlADMM compatible with convolutional networks was not available. Time-axis is logscale. The diamonds mark the end of the first and last epoch for each method. Refer to Appendix A.7.2 for experimental details.

that in practice, our implementations of LFP rely on the LXT library (Achtibat et al., 2024), which simply overwrites the forward and backward passes and follows the above computational complexity. We also implement LFP for the zennit library (Anders et al., 2021), which utilizes a version of gradient checkpointing for increased memory efficiency, but in turn requires two forward passes and one backward pass per batch update.

To analyze computational overhead associated with LFP, we compare its performance and required clock time against stochastic gradient descent, feedback alignment (Lillicrap et al., 2016), dlADMM (Wang et al., 2019), and a variant of Particle Swarm Optimization (PSO) (Eberhart & Kennedy, 1995). We chose these methods based on their applicability to standard DNNs, distinctness in optimization strategy, and availability of implementation. Refer to Appendix A.7.2 for details on each method's hyperparamters. We train a simple MLP and LeNet for 50 epochs on MNIST using the above approaches, if applicable. As shown in Figure 5, LFP has computational overhead comparable to e.g. gradient descent and feedback alignment, which also perform a backward pass.

### 3.5 LFP for non-ReLU Activations

We have shown in Section 3.3, that LFP-0, as formulated in Equation 4, is equal to weight-scaled gradient descent, but may differ in the magnitude of updates. This does not hold for models whose activation functions are not partially linear (as well as for the LFP-$\varepsilon$ rule applied in practice), in which the updates caused by LFP and gradient descent may differ more significantly (cf. also Appendix A.4). Consequently, we investigate empirically in the following Section whether LFP generates meaningful update signals when applied to such models.

Table 3 compares performances of LFP-trained models across various activation functions (ReLU, SiLU, ELU, Tanh, Sigmoid, and Heaviside Step Function, from left to right). These activations are further illustrated in Figure A.11. Note that among these, LFP-0 would equal weight-scaled gradient descent only for ReLU activations. Refer to Appendix A.7.3 for further experimental details.

We observe that LFP on ReLU-activated models outperforms all other activations (except for the VGG-like model, where ELU wins instead). However, with SiLU, ELU and Tanh activations LFP generally also converges, albeit with slightly lower performance — depending on the model and data. This is interesting, as there is no equivalence between LFP and gradient descent for these activations (cf. Theorem 1, which relies on a subset of piece-wise linear activations), but LFP still arrives at a good solution.

For the remaining activation functions, Heaviside and Sigmoid, LFP leads to (close-to) randomly performing models in this experiment[8]. With LFP being applicable to Heaviside-activated models in theory, this result is initially surprising,

---

[8]Note that this random baseline varies strongly between ResNet-18 and VGG-16 for the ISIC2019 dataset. This dataset contains 9 extremely imbalanced classes, with one class "NV" containing approximately half the total amount of samples. Depending on which class is favored, the accuracy of randomly performing models can thus vary significantly.

Table 3: Accuracy of LFP with different activation functions. Mean and standard deviation over three seeds were evaluated. For all models except VGG-like, LFP on ReLU-models outperforms (underlined) all other activations. However, for several not piece-wise linear functions (SiLU, ELU, Tanh) LFP achieves accuracies that are close to that, showing that LFP is also able to train models with not partially linear activation functions. For reference, we also provide gradient descent results in *grey*.

| Model | Data | Activation | | | | | |
|-------|------|------|------|------|------|---------|-----------|
| | | ReLU | SiLU | ELU | Tanh | Sigmoid | Heaviside[9] |
| VGG-like | Cifar10 | 82.2±0.4 | 77.1±0.6 | 82.7±0.4 | 78.8±0.2 | 10.0±0.0 | 10.0±0.0 |
| | | *84.1±0.4* | *85.3±0.3* | *85.2±0.3* | *80.7±0.1* | *10.0±0.0* | *16.0±0.5* |
| | Cifar100 | 49.0±0.6 | 45.7±0.6 | 51.0±0.3 | 20.6±0.6 | 1.0±0.0 | 1.0±0.0 |
| | | *52.0±0.6* | *55.8±0.6* | *55.5±0.4* | *51.1±0.3* | *1.0±0.0* | *2.4±0.1* |
| VGG-16 | CUB | 72.9±0.2 | 72.6±0.5 | 0.5±0.0 | 1.0±0.9 | 0.5±0.0 | 0.5±0.0 |
| | | *71.6±0.1* | *71.7±0.3* | *69.9±0.4* | *5.3±3.9* | *0.5±0.0* | *2.8±0.1* |
| | ISIC2019 | 86.5±0.2 | 84.8±0.4 | 80.0±0.6 | 50.6±0.0 | 50.6±0.0 | 50.6±0.0 |
| | | *87.1±0.2* | *86.9±0.1* | *86.9±0.3* | *50.6±0.0* | *50.6±0.0* | *54.5±0.1* |
| | Food-11 | 91.7±0.2 | 91.2±0.5 | 91.0±0.4 | 30.9±15.4 | 14.9±0.0 | 14.9±0.0 |
| | | *91.9±0.2* | *91.2±0.0* | *90.7±0.1* | *31.3±23.2* | *14.9±0.0* | *33.0±0.2* |
| ResNet-18 | CUB | 66.7±0.3 | 54.2±0.2 | 47.4±0.3 | 12.4±1.5 | 0.5±0.0 | 0.5±0.0 |
| | | *70.4±0.3* | *63.5±0.6* | *47.9±0.3* | *16.5±0.3* | *3.7±0.1* | *1.5±0.1* |
| | ISIC2019 | 85.1±0.6 | 80.3±0.2 | 77.3±0.5 | 72.9±0.7 | 17.7±0.0 | 17.7±0.0 |
| | | *86.0±0.6* | *84.8±0.2* | *82.4±0.3* | *78.8±0.3* | *69.5±0.3* | *54.5±0.3* |
| | Food-11 | 90.9±0.5 | 86.8±0.5 | 82.4±0.4 | 76.0±0.7 | 11.0±0.0 | 11.0±0.0 |
| | | *91.7±0.1* | *88.7±0.3* | *86.4±0.4* | *82.2±0.3* | *71.2±0.5* | *29.8±0.2* |

especially combined with the above-random performance of gradient descent. We hypothesize that such models may be difficult to train in practice, since the Heaviside function has no decreasing slope before evaluating to zero, and thus, in classical ANN architectures, leads more easily to dead neurons than other, comparable activations, e.g., ReLU. We investigate this effect more closely in Section 3.6 and provide a solution for LFP. We further show that Heaviside activations cause no issues for LFP when a different model type is used, i.e., in the context of SNNs in Section 4.2, which accumulate stimuli over time and thus may be more robust to dead neurons.

Sigmoid, on the other hand, is equal to a scaled and shifted Tanh function. As LFP is generally able to train Tanh-activated models, this illuminates a requirement of LFP: Activation functions need to map positive values to $[0, \infty]$ and negative values to $[-\infty, 0]$. Otherwise, pre-activation and post-activation may be differently signed, and the accumulated (post-activation) reward (cf. Equation 3, $r_i$) cannot be used directly as the (pre-activation) reward (cf. Equation 2, $r_j$) in order to further backpropagate feedback[10]. We note, however, that this requirement is rarely not fulfilled in state-of-the-art deep learning models; consequently, it only impacts the applicability of LFP to non-deep-learning machine learning applications in practice. The Tanh activation fulfills this requirement, but Sigmoid does not. In comparison, for gradient-based training, fulfilling this property is not a strict requirement, but nevertheless advantageous, as evidenced by the large performance gap between Tanh and Sigmoid activated models (Sigmoid-activated models achieve above-random performance only for ResNet-18) for the gradient-descent baseline.

In general, we observe that the optimal configurations (i.e., underlined values in Table 3) for gradient-descent slightly outperform the optimal configurations of LFP. This difference varies between models and datasets, but exceeds 1% only in three settings. We attribute this performance gap to the additional implicit weight-scaling of LFP, which may impact accuracy slightly negatively, but provide advantages in terms of model sparsity (cf. Section 4.1). In summary, we find that LFP is able to train models whose activation functions are not partially linear; indicating that it functions as a training paradigm in settings where it is not equal to weight-scaled gradient descent (also cf. Section 4.2, where the condition on activations from Theorem 1 is also not fulfilled).

---

[9]Note that we investigate the reason behind LFP failing to train Heaviside-activated DNNs in Section 3.6, provide a solution, and demonstrate its efficacy in Heaviside-activated SNN (cf. Section 4.2).

[10]If pre-activation and post-activation have different signs, the LFP formulas could be adapted heuristically by simply taking the oppositely signed post-activation reward as pre-activation reward

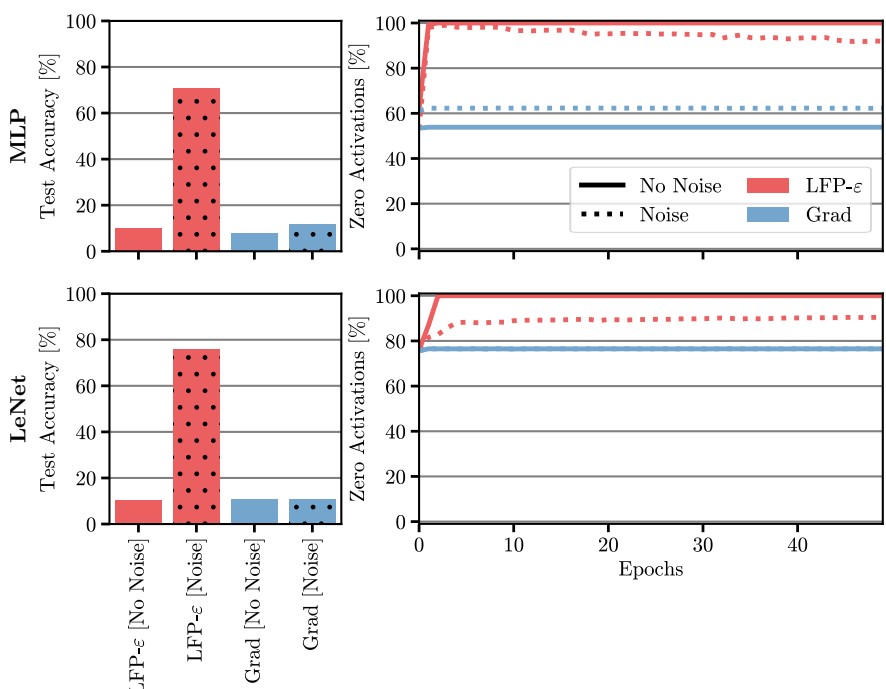

Figure 6: Effect of adding noise to activations during training on LFP and gradient descent performance on Heaviside-activated DNNs. *(Top row)*: Small MLP; *(Bottom row)*: LeNet. We report the test accuracy of the final model *(left)* and the percentage of zero activations (averaged over all batches within an epoch) after the last Heaviside for the first 50 epochs *(right)*. Models with noise applied to activations during training are shown as dotted hatch and line, respectively. Adding noise to activations during training helps (only) LFP, vastly increasing the achieved test performance.

## 3.6 Training Heaviside-activated DNNs with LFP

Although propagating rewards through Heaviside activations with LFP is feasible in theory, as LFP does not require the computation of gradients, the results presented in Table 3 show a negative result. This finding is initially unexpected, particularly in comparison to the gradient descent trained baseline models, which perform above random. Note, however, that the models evaluated in Table 3 were pre-trained on ImageNet using gradient descent and employ a linear activation in the final layer. It is therefore reasonable to expect that gradient descent would achieve above random performance in these tasks by leveraging the pre-trained lower-layer parameters while updating only the final layer.

For LFP, we hypothesize that it fails to effectively train the Heaviside-activated DNNs in this experiment due to the emergence of a substantial number of dead (i.e., always zero-activating) neurons during the initial training phase. Since such neurons neither contribute nor propagate rewards backward, they effectively obstruct both forward and backward information flow, thereby preventing the updating of lower layers. As discussed previously, this property makes the algorithm more efficient and realizes a property of biological learning (Pozzi et al., 2020). We further observe that LFP successfully trains the SNNs discussed in Section 4.2, despite their reliance on Heaviside activations. We attribute this discrepancy to the noisy inputs and dynamic, time-dependent nature of SNNs that accumulate membrane potential over time.

If this hypothesis is correct, an initial increase in inactive neurons should be observable when training Heaviside-activated DNNs with LFP, but no such change should occur with gradient descent due to the zero-derivatives of the Heaviside function. Furthermore, introducing a small degree of noise to activations during training (there seems to be evidence for noisiness in biological neurons as well (Buhmann & Schulten, 1987)) should mitigate the issue of inactive neurons, thereby enabling LFP to produce meaningful, albeit slightly noisy, updates. To evaluate this hypothesis, we conduct experiments on the MNIST dataset, as illustrated in Figure 6. We test a three-layer MLP with full Heaviside activations, including the final layer, and a LeNet with Heaviside activations in all layers except the last. The MLP comprises linear layers with 120, 84, and 10 neurons, respectively. The LeNet architecture consists of two convolutional

layers with 16 channels and a kernel size of 5, each followed by max-pooling with a stride of 2. Two subsequent linear layers contain 120, 84, and 10 neurons for classification. Both models are trained with a learning rate of 0.1 and a batch size of 128 for 300 and 400 epochs, respectively.

As shown in Figure 6, a rapid increase in inactive neurons occurs within the first few iterations when using LFP, whereas no such effect can be observed for gradient descent. Moreover, introducing minor noise to activations effectively mitigates this issue for LFP, leading to vastly improved test accuracy in the final Heaviside-activated model (Note that no noise is applied during testing). Furthermore, despite the final layer of LeNet being linearly activated, gradient descent achieves only random performance on that model. This suggests that the above-random performance of gradient descent in Table 3 can be attributed to the advantageous pre-trained weights in conjunction with training of the linearly activated final layer.

Note, however, that a limitation of this noise-based approach is the increased training time, requiring 300 epochs for the MLP and 400 epochs for the LeNet. Different noise, or a more informed "drop-in" strategy may be able to resolve this, however, we leave investigation into this to future work.

## 4 Applications

In the following, we will consider two specific applications for LFP, based on the above observations:

### 4.1 Application 1: Sparse and Prunable Models

In Section 3.3, we observed that in the investigated toy settings, LFP seems to lead to sparser models than gradient descent, i.e., the resulting models have few comparatively large weights and many weights that are close to zero. Sparser models utilize the available parameters more efficiently, in the sense that a larger number of connections can be pruned (i.e., removed) before performance deteriorates, allowing for reduced memory requirements when storing the model. In this Section, we explore this model sparsity property of LFP in more detail. Refer to Appendix A.7.4 for details on the experimental setup.

For this purpose, we first confirm that our observations regarding weight distribution from Section 3.3 extend to larger scale models as well. The layer-wise normalized weight distribution for LFP- and gradient-trained ResNet-18 models is visualized in Figure 7 *(top)*. The *bottom* plot shows the average Gini coefficient (Hurley & Rickard, 2009) of the visualized distribution at each layer, as a measure of sparsity. Note that we canonized (see Appendix A.7.4) the BatchNorm layers of the ResNet into the preceding convolutional layers to account for their implicit re-scaling of weight magnitudes. Results for more models are available in Figure A.6. We find that indeed, compared to gradient-trained models, LFP-trained models are sparser, with more weights close to zero and few large weights, as well as higher Gini indices. I.e., relative to the respective weight magnitude in each layer, the visualized percentiles up to the 75th are lower for LFP than for gradient descent.

However, the fact that the weight distribution of an LFP-trained model is closer to zero does not necessarily imply that its weights are sparser, in the sense that less weights are relevant. We therefore study further how this changed weight distribution affects the prunability of the resulting models. Here, we first employ a magnitude-based pruning criterion akin to Han et al. (2015). If a model retains performance for longer while being pruned under this criterion, it either has a better connection between weight magnitude and weight importance or is practically sparser, with more weights being safely removable. To test whether the latter case applies, we thus additionally employ a relevance-based pruning criterion (Voita et al., 2019; Yeom et al., 2021; Hatefi et al., 2024). The pruning criteria are described in more detail in Appendix A.7.4. The results of this analysis are visualized in Figure 8. LFP-trained models generally retain performance better than gradient-trained ones under magnitude-based local pruning (*left*) and global pruning (*middle*), i.e., the curve only starts decreasing at a higher pruning rate. This implies that both within each layer, and globally across the whole model, there is a better weight magnitude-importance connection, or that more weights can be removed from the resulting model. Since LFP-trained models also perform better under the relevance pruning criterion (*right*), the latter is the case. Results for additional models and datasets are available in Figure A.9. Note that we observed the final LFP-trained models to perform slightly worse than gradient-trained models during this experiment (cf. Figure A.9), which we attribute to slightly slower convergence due to the increased sparsity. Furthermore, there are certain few combinations of dataset, model, and pruning criterion, where gradient-trained models outperform the LFP-trained models in terms of prunability (for instance, CUB, ResNet-34, and relevance-based pruning). Interestingly, these

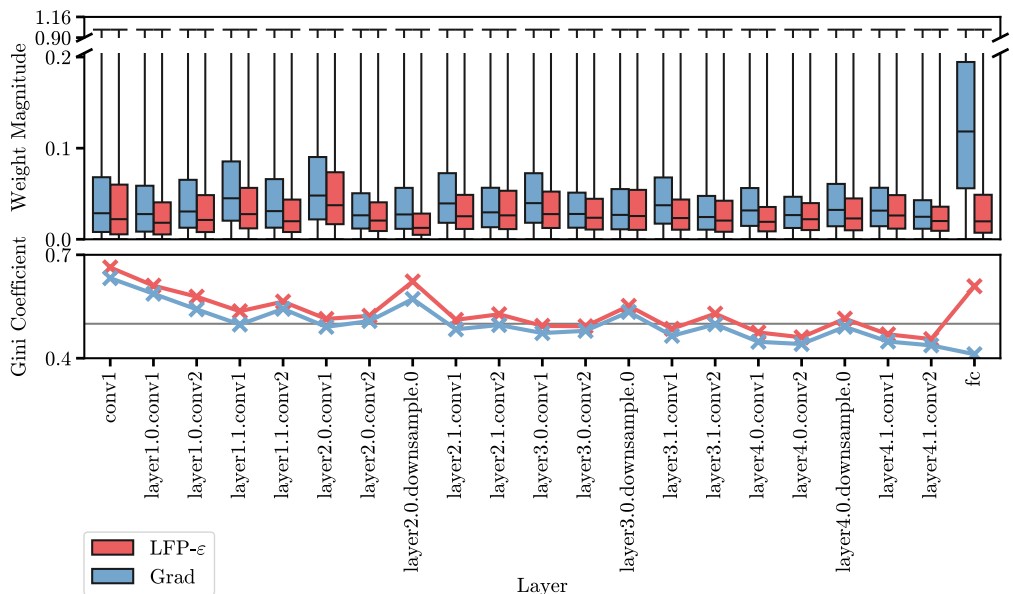

Figure 7: Sparsity of LFP-trained models vs. gradient-trained models. *Top*: Distribution of weight magnitudes, normalized by the maximum absolute value per layer. Data is shown for ResNet-18 models, trained on the CUB dataset using either LFP or gradient descent (Grad). BatchNorm layers were canonized into the preceding convolutional layers. Black horizontal lines of each bar denote, from the bottom to the top, the 0-th, 25th, 50th, 75th, and 100th percentile of weight magnitudes in each layer. *Bottom*: Gini coefficient of normalized weights at each layer, as a measure of sparsity. A higher Gini coefficient implies higher sparsity. Relative to the layer-wise maximum magnitude, LFP-trained models have a larger amount of smaller weights close to zero in each layer, indicating increased sparsity compared to the gradient-trained models. This is confirmed by the Gini coefficient, where the score of the LFP-trained models is larger or equal to that of gradient-trained models. Results are averaged over three seeds. Refer to Figure A.7 for unnormalized weight distributions, and Figure A.6 for weight distributions of additional models.

cases are restricted to magnitude-based local pruning and relevance-based global pruning. Nevertheless, LFP-trained models seem to retain performance better in the vast majority of settings. Also note that, since LFP is equivalent to weight-scaled gradient descent in ReLU models (cf. Theorem 1) the above results also align with the observations of Kim et al. (2020).

In combination, the above results demonstrate how LFP results in increased network sparsity as the model is incentivized to utilize the available connections efficiently. LFP-trained models are thus more resistant against pruning, and can retain performance even when a larger amount of parameters is removed.

## 4.2  Application 2: Gradient-Free Training

As discussed previously, LFP functions as a gradient-free learning algorithm, which makes it readily applicable to models that contain non-differentiable components. Consequently, LFP is well-suited for training a broader range of models than those trainable through gradient-based methods, including some overlap of models that both methods can be applied to.

In this section, we demonstrate the versatility of LFP by applying it to train a Spiking Neural Network (SNN), a neuromorphic recurrent neural network architecture. SNNs process information in the form of binary signal sequences known as spikes, mimicking the functionality of biological neurons. When implemented on specialized neuromorphic hardware, SNNs enable event-based and energy-efficient processing while requiring less memory compared to conventional network architectures (Maass, 1997; Ponulak & Kasinski, 2011). However, the binary activation functions in SNNs present a challenge: their gradients are either undefined or zero, rendering direct application of gradient-based learning methods infeasible. In practice, this issue can be mitigated by replacing the network during the backward pass with a differentiable approximation. Such approximations, however, could be *extremely inefficient* when implemented

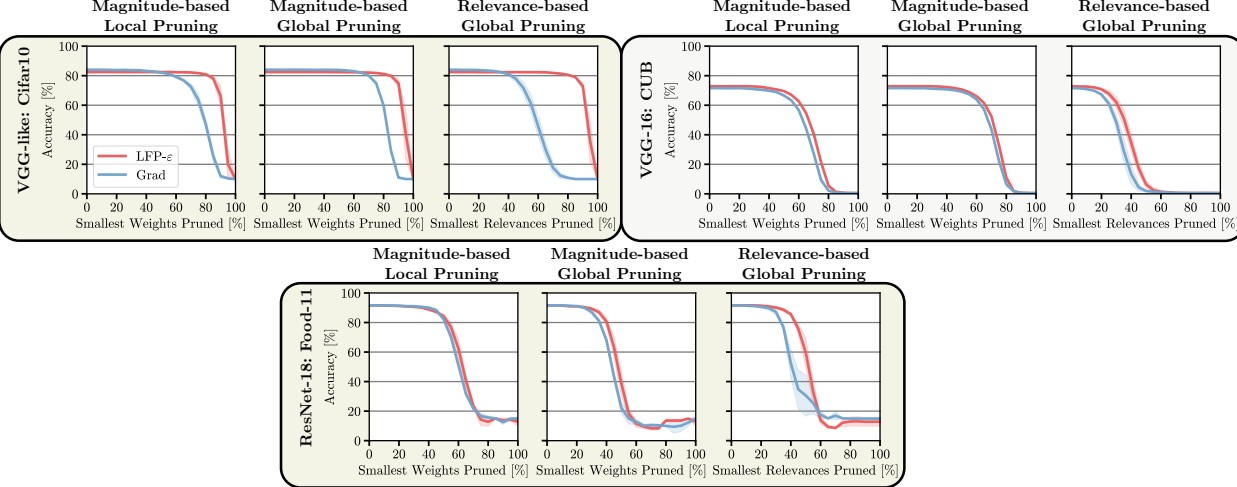

Figure 8: Test set accuracy over the percentage of pruned weights. BatchNorm layers were canonized into preceding convolutional layers prior to pruning (cf. Appendix A.7.4). Results are shown for three pruning criteria (*left to right:* pruning weights with the smallest magnitude within each layer, pruning weights with the smallest magnitude globally, and pruning weights with the smallest relevance globally). LFP-trained models are able to retain performance for longer when pruned based on weight magnitude, both locally and globally. Interestingly, LFP-trained models also retain performance better under the relevance-based pruning criterion, indicating that the resulting models are indeed sparser and use fewer weights to solve a given task. Results are averaged over three seeds, with the shaded area showing the standard deviation across seeds. Refer to Figure A.9 for results on additional models and datasets, and to Appendix A.7.4 for more details w.r.t. the pruning algorithms.

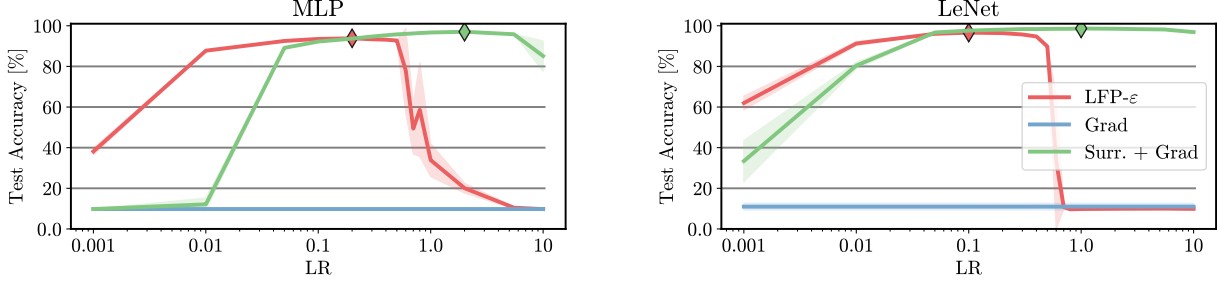

Figure 9: Final test accuracies after three epochs of training with different learning rates. *Left:* SNN based on a fully-connected MLP architecture, *right:* SNN with a LeNet backbone. The diamond marks the highest performance. LFP achieves high accuracies across a wide range of learning rates. The top performance of LFP-$\varepsilon$ and surrogate-enabled gradient descent (Surr.+Grad) are comparable, where the latter yields a slightly higher top accuracy. Direct application of gradient descent (Grad) does not yield any improvement at all (since learning here is not possible as $\frac{dw}{dL} = 0$). More details on the hyperparameters and setup are provided in Appendix A.7.5. Results are averaged over three seeds, with the shaded area showing the standard deviation across seeds.

directly in hardware, as the (binary) activation functions would need to be replaced by continuous surrogates (Pfeiffer & Pfeil, 2018). LFP offers a training paradigm for SNNs that overcomes this limitation by propagating rewards directly through the non-differentiable spiking activations.

To demonstrate the effectiveness of LFP, we trained two types of SNN architectures on the MNIST handwritten digit classification task: a fully-connected network and a convolutional LeNet architecture. See Appendix A.7.5 for further details. For comparison, we also trained both architectures using surrogate-enabled gradient descent.

Figure 9 displays the final test set accuracies obtained for various learning rates. We find that LFP-$\varepsilon$-based training achieves high accuracies across a wide range of learning rates. The highest accuracies obtained via LFP and surrogate-

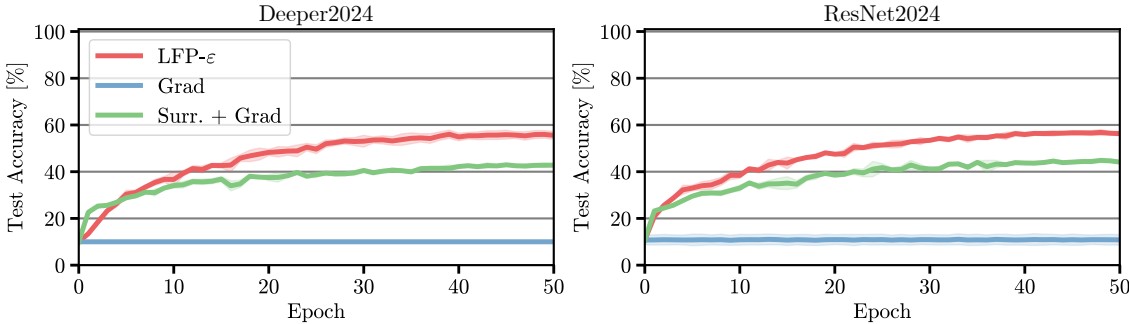

Figure 10: Test accuracies over epochs of training two larger SNN-architectures on Cifar10 data. Architectures are inspired by the equally named ones from `https://github.com/aidinattar/snn`. Results are plotted for the learning rate which yielded the best performing run amongst the learning rates we investigated. LFP outperforms both gradient descent and surrogate gradient descent after a few epochs. More details on the hyperparameters and setup are provided in Appendix A.7.5. Results are averaged over three seeds, with the shaded area showing the standard deviation across seeds.

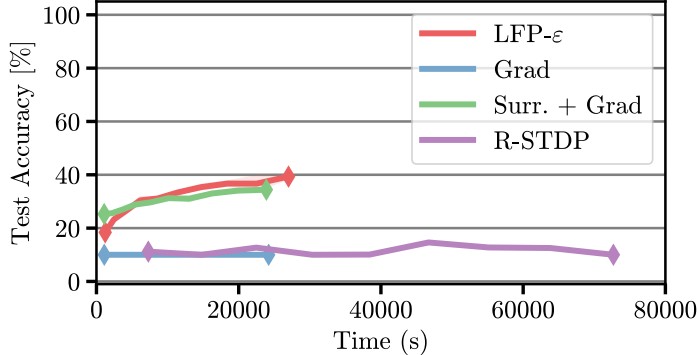

Figure 11: Evaluation of train accuracy trajectories for the Deeper2024 SNN trained on Cifar10 with LFP, R-STDP, and gradient-based training enabled by substantial model augmentation. It is important to note that R-STDP sequentially trains each layer, thus train accuracy metrics only become available upon reaching the final layer, which occurs after approximately 7500 seconds. Due to the large training time of R-STDP we report results only for ten epochs in this figure. While both LFP and surrogate gradient descent achieve similar performance levels (which increase even further with additional epochs, cf. Figure 11), R-STDP performance remains close to random. Refer to Appendix A.7.5 for experimental details.

enabled gradient-based training are comparable, albeit LFP results in a slightly lower top accuracy for both trained architectures. Consistent with the experiments in Section 3.3, we observe that LFP performance saturates at lower learning rates compared to gradient-based training. Also for other hyperparameters, such as the sequence length used to encode data and the decay rate of the internal state of the spiking neurons (see Appendix A.6), LFP and gradient descent show different sensitivities. Specifically, for the LeNet architecture, LFP performs worse than surrogate-enabled gradient descent when encoding the training data with smaller sequence lengths, while it performs better for higher decay rates of the internal neuron states, see Figure A.10 in the Appendix.

We further extend this experiment by training two larger convolutional SNN architectures on Cifar10 data (Figure 10). Notably, LFP outperforms both gradient descent and vanilla gradient descent after a few epochs on both models. We also applied LFP with surrogate in the backward pass as a sanity check, which yielded the exact same curve as LFP without surrogate and is thus not visualized in the figure.

In addition to performance, we compare the training efficiency of different methods. Here, we additionally apply R-STDP (Mozafari et al., 2018), an alternative gradient-free learning framework for SNNs that independently trains each layer based solely on its respective inputs and outputs, as discussed in Section 2. The findings are illustrated

in Figure 11. LFP has similar time complexity to gradient descent and is faster than R-STDP. In this setting, LFP outperforms both gradient descent and R-STDP in terms of accuracy (also cf. Figure 10).

Overall, our results show that LFP-based training of SNNs efficiently achieves high accuracies comparable to (or even exceeding) those obtained via surrogate-enabled gradient-based training for different architectures across various hyperparameters and different underlying architectures. This makes LFP a promising alternative for SNN training, especially in hardware implementations where surrogate gradients might be impractical.

# 5 Discussion

In this work, we introduce LFP, a novel XAI-based paradigm for training neural networks that decomposes a *global reward* signal into neuron-wise *local rewards* to update parameters without requiring the computation of gradients.

We show empirically and theoretically, that in supervised classification settings, LFP converges under the right choice of hyperparameters. While our convergence proof relies on models containing piecewise linear activations, we demonstrate experimentally that LFP also converges for other activations, such as SiLU, ELU, and Tanh. We further identify several strengths of LFP, among these, the increased sparsity of information representation in the resulting model, and the independence from the differentiability of the model or objective. Based on these properties, we investigate two applications: The increased prunability of LFP-trained models due to their sparsity and the approximation-free training of SNNs. In the former setting, we show how LFP-trained models use fewer weights to solve a given task and retain better performance at higher pruning rates compared to gradient-descent-trained models. In the latter setting, LFP is able to train SNNs with no performance trade-off compared to (approximative) gradient descent, but without requiring approximation of the Heaviside step activations during the backward pass. In summary, we provide a proof-of-concept for the novel LFP training paradigm, observing that the method has several applications and — with future research beyond this work — further potential to develop.

## 5.1 Strengths and Limitations of LFP

Consequently, we summarize the strengths and limitations of LFP as follows:

**Strengths.** Firstly, LFP can utilize both differentiable and non-differentiable objective functions. While differentiable objectives as employed e.g. for supervised gradient descent can be reformulated as rewards for LFP, the reverse is not always the case. Despite this, LFP is closely related to gradient descent, with similar computational efficiency and convergence properties, but requires no gradient computation and can thus train non-differentiable models. This includes, e.g., Heaviside step function-activated SNNs, where LFP converges to similar (or even better) performances as (approximative) gradient descent but does not require approximation in the backward pass, easing the adoption of neuromorphic hardware. Secondly, LFP finds sparse solutions without compromising accuracy. The resulting models consequently retain performance better under pruning. In combinations, these properties have promising implications towards utilizing LFP for energy-efficient AI in future work — especially in state-of-the-art applications such as deploying Large-Language models in energy- and memory-restricted applications.

**Limitations.** Nevertheless, there are several limitations to the applicability of LFP: Firstly, it assumes that the model consists of neurons $\phi(h(a))$ where $h(a)$ is a linear function of $a$ and $\phi$ an optional nonlinearity. This is the case, e.g., for neural networks, but does not include the same wide range of functions optimizable by gradients. Nevertheless, this assumption is met by deep learning applications, where LFP is best suited. Secondly, while activations do not need to be differentiable, they do need map positive values to $[0, \infty]$ and negative values to $[-\infty, 0]$, although this requirement is usually satisfied by deep learning models in practice. Thirdly, connections that do not contribute in the forward pass will not be updated and will not propagate feedback backward. This originates from the LFP assumption of zero as a neutral value, inherited from LRP. Consequently, the backward pass through and update of a connection is blocked by either the corresponding activation or parameter being zero (cf. gradient descent, where backward passes are only blocked through zero parameters at a connection and, vice versa, parameter updates are only blocked through zero activations, offering more options to escape dead neurons). Note, however, that this behavior is closer to the biological reality (Pozzi et al., 2020), enables sparse and thereby energy-efficient updates, and — if desired — can largely be mitigated in practice by utilizing a momentum term or drop-in strategies such as adding noise to activations in the forward pass. Finally, the proposed reward propagation through BatchNorm layers (cf. Section 3.1.4) is a novel contribution and needs further testing in different models and in company with different LFP rules. However, in our experiments, models

that contained BatchNorm layers reached comparable accuracies in our experiments whether they were trained by LFP or gradient descent, which makes the rule seem promising for LFP- and LRP-applications alike.

## 5.2 Future Work

This paper focuses on introducing Layer-wise Feedback Propagation, demonstrating its convergence, and suggesting some applications of this novel training method. For LRP, which LFP leverages to distribute rewards, there exist several rules specific to explaining certain types of models or layers. Not all of these rules may translate well to LFP (due to the diverging goals of explaining vs. training, cf. Appendix A.5), or they might require non-trivial adaptations. Further LFP-rules for BatchNorm layers and transformer architectures (cf. Achtibat et al. (2024) and Appendix A.9) nevertheless seem worthwhile to explore in follow-up works, as they would enable the application of LFP to a wider range of architectures and state-of-the-art settings. In this work, we utilize LRP to obtain local rewards for the update in Equation 4. Exploration into alternative sources for this local reward seems promising, e.g. ones that do not rely on a backward pass, especially for training efficiency in SNN-applications. Similarly, we investigate LFP in-depth for supervised classification in this work. In the future, exploration of other learning problems (such as regression, cf. Appendix A.8) and comparison with a larger number of training techniques would be interesting, especially in the context of reinforcement learning, which LFP is related to. We furthermore leave an in-depth investigation of additional hyperparameters, e.g., the choice of $\varepsilon$, or thorough exploration of potential reward functions to future research.

## Acknowledgments

🇪🇺 This work was supported by the European Union's Horizon Europe research and innovation programme (EU Horizon Europe) as grants [ACHILLES (101189689), TEMA (101093003)]; and the European Union's Horizon 2020 research and innovation programme (EU Horizon 2020) as grant iToBoS (965221).

This work was further supported by the Federal Ministry of Education and Research (BMBF) as grant BIFOLD (01IS18025A, 01IS180371I) and the German Research Foundation (DFG) as research unit DeSBi [KI-FOR 5363] (459422098). This research / project is supported by the National Research Foundation, Singapore and Infocomm Media Development Authority under its Trust Tech Funding Initiative. Any opinions, findings and conclusions or recommendations expressed in this material are those of the author(s) and do not reflect the views of National Research Foundation, Singapore and Infocomm Media Development Authority.

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

# A  Appendix

## A.1  Proof-Of-Concept with Additional Activations

In Section 3.3 we demonstrated the convergence of LFP for ReLU-activated models in toy settings. Here, we investigate the efficacy of LFP for training models with other activations — of which several are not piece-wise linear and thus not covered under Theorem 1.

We use the same experimental setup as for Figure 3 (also cf. Section A.7.1), but insert an additional activation after the last linear layer. Otherwise, due to the simplicity of the toy data, a learning algorithm may solve the given tasks while ignoring the previous layers, and thus bypass the activation functions that we vary in this experiment.

Results are shown in Figure A.1. LFP generally results in the expected decision boundaries. Nevertheless, there is some variation between activation functions. Specifically for Heaviside, while the test accuracy is clearly better than random, and the model has clearly been somewhat optimized to solve the task, accuracy and resulting boundaries are considerably worse compared to other activation functions. Furthermore, during training, we observed a high dependence on initialization for Heaviside-activated models — even in the simple toy settings considered here. We hypothesize that while the update signals are sensible, Heaviside activations make standard neural network training simply too unstable (but this may not be the case for other types of architectures). This corresponds well to our results in Sections 3.5, 3.6, and 4.2.

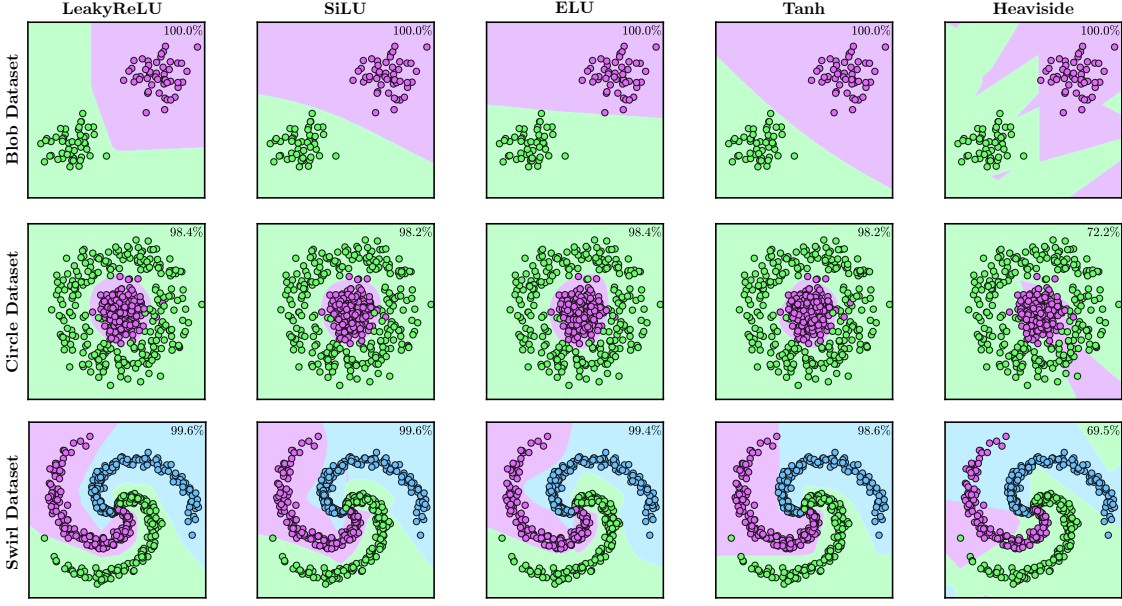

Figure A.1: Additional decision boundaries for the three toy datasets from Figure 3 with other, non-ReLU activation functions (also cf. Figure A.11). The test accuracy of the corresponding model is shown in the upper right corner of each plot. LFP is able to train models with all of these activations - not only the piece-wise linear ones covered by Theorem 1.

## A.2 Proof of Theorem 1

*Proof.* In the interest of preciseness, let $a_i^{[l]}$ denote the $i$-th neuron in the $l$-th layer of the network and $w_{ij}^{[l+1]}$ the weight corresponding to the connection of neuron $a_i^{[l]}$ to $a_j^{[l+1]}$. Similarly, let $r_i^{[l]}$ denote the reward corresponding to neuron $a_i^{[l]}$ and $r_{ij}^{[l+1]}$ the reward corresponding to the weight at the connection of neuron $a_i^{[l]}$ to $a_j^{[l+1]}$. We show that the equation

$$r_j^{[l]} = a_j^{[l]} \sum_c \frac{do_c}{da_j^{[l]}} \frac{r_c}{o_c} \tag{A.1}$$

holds for all hidden layers of the network by inverse induction from the last layer before the output $l = L$ to the first layer $l = 1$. For calculating the output activations $o_c$, we use a linear activation function s.t. $o_c = z_c^{[L+1]} = \sum_j a_j^{[L]} w_{jc}^{[L+1]}$. Hence

$$a_j^{[L]} \sum_c \frac{do_c}{da_j^{[L]}} \frac{r_c}{o_c} = \sum_c a_j^{[L]} \frac{d\left(\sum_j a_j^{[L]} w_{jc}^{[L+1]}\right)}{da_j^{[L]}} \frac{r_c}{o_c}$$

$$= \sum_c a_j^{[L]} w_{jc}^{[L+1]} \frac{r_c}{z_c^{[L+1]}} = \sum_c \frac{z_{jc}^{[L+1]}}{z_c^{[L+1]}} r_c = r_j^{[L]}.$$

For the induction step, assume now that the equality in Equation A.1 holds for layer $l + 1$. The following equation follows from the formula for relevance propagation:

$$r_i^{[l]} = \sum_j \frac{z_{ij}^{[l+1]}}{z_j^{[l+1]}} r_j^{[l+1]} = \sum_j \frac{a_i^{[l]} w_{ij}^{[l+1]}}{z_j^{[l+1]}} \left\{ a_j^{[l+1]} \sum_c \frac{do_c}{da_j^{[l+1]}} \frac{r_c}{o_c} \right\}$$

$$= a_i^{[l]} \sum_j \frac{\mathrm{ReLU}(z_j^{[l+1]})}{z_j^{[l+1]}} w_{ij}^{[l+1]} \left\{ \sum_c \frac{do_c}{da_j^{[l+1]}} \frac{r_c}{o_c} \right\}$$

$$= a_i^{[l]} \sum_j \mathbb{1}(z_j^{[l+1]} > 0) w_{ij}^{[l+1]} \left\{ \sum_c \frac{do_c}{da_j^{[l+1]}} \frac{r_c}{o_c} \right\}$$

$$= a_i^{[l]} \sum_j \frac{da_j^{[l+1]}}{da_i^{[l]}} \left\{ \sum_c \frac{do_c}{da_j^{[l+1]}} \frac{r_c}{o_c} \right\} = a_i^{[l]} \sum_c \frac{do_c}{da_i^{[l]}} \frac{r_c}{o_c}$$

Thus by induction Equation A.1 holds for all hidden layers $l$. We apply Equation A.1 to the formula for the reward $r_{ij}^{[l+1]}$:

$$r_{ij}^{[l+1]} = \frac{z_{ij}^{[l+1]}}{z_j^{[l+1]}} r_j^{[l+1]} = \frac{w_{ij}^{[l+1]} a_i^{[l]}}{z_j^{[l+1]}} \mathrm{ReLU}(z_j^{[l+1]}) \sum_c \frac{do_c}{da_j^{[l+1]}} \frac{r_c}{o_c}$$

$$= w_{ij}^{[l+1]} a_i^{[l]} \mathbb{1}(z_j^{[l+1]} > 0) \sum_c \frac{do_c}{da_j^{[l+1]}} \frac{r_c}{o_c}$$

$$= w_{ij}^{[l+1]} \frac{da_j^{[l+1]}}{dw_{ij}^{[l+1]}} \sum_c \frac{do_c}{da_j^{[l+1]}} \frac{r_c}{o_c}$$

$$= w_{ij}^{[l+1]} \sum_c \frac{do_c}{dw_{ij}^{[l+1]}} \frac{r_c}{o_c}$$

Inserting the reward defined in the statement of the theorem, we derive the following form of the reward:

$$r_{ij}^{[l+1]} = w_{ij}^{[l+1]} \sum_c \frac{do_c}{dw_{ij}^{[l+1]}} \frac{o_c \cdot \left(-\frac{d\mathcal{L}}{do_c}\right)}{o_c}$$

$$= -w_{ij}^{[l+1]} \frac{d\mathcal{L}}{dw_{ij}^{[l+1]}}$$

The update step of LFP in Equation 4 is given by

$$d_{w_{ij}^{[l+1]}}^{\text{lfp}} = \frac{|w_{ij}^{[l+1]}| \cdot a_i^{[l]}}{z_j^{[l+1]}} \cdot r_j^{[l+1]}$$

$$= \frac{|w_{ij}^{[l+1]}|}{w_{ij}^{[l+1]}} \frac{z_{ij}^{[l+1]}}{z_j^{[l+1]}} \cdot r_j^{[l+1]} = \frac{|w_{ij}^{[l+1]}|}{w_{ij}^{[l+1]}} \cdot r_{ij}^{[l+1]},$$

so the update is equivalent to weight-scaled gradient descent:

$$w_{ij}^{[l+1]\,\text{new}} = w_{ij}^{[l+1]} + \eta \cdot d_{w_{ij}^{[l+1]}}^{\text{lfp}}$$

$$= w_{ij}^{[l+1]} + \eta \cdot \frac{|w_{ij}^{[l+1]}|}{w_{ij}^{[l+1]}} \cdot r_{ij}^{[l+1]}$$

$$= w_{ij}^{[l+1]} + \eta \cdot \frac{|w_{ij}^{[l+1]}|}{w_{ij}^{[l+1]}} \left\{ -w_{ij}^{[l+1]} \frac{d\mathcal{L}}{dw_{ij}^{[l+1]}} \right\}$$

$$= w_{ij}^{[l+1]} - \eta \cdot |w_{ij}^{[l+1]}| \frac{d\mathcal{L}}{dw_{ij}^{[l+1]}}$$

$\square$

Note that while we proved the statement for ReLU functions with linearly activated outputs, it in fact holds that for weight-scaled gradient descent with any activation function $\phi_i^l(z_i^l) = \frac{d\phi_i^l(z_i^l)}{dz_i^l} \cdot z_i^l$, Theorem 1 yields an equivalent LFP-0-reward with equivalent weight updates.

### A.3  LFP with Reward from Equation 17

While we observed reward #3 from Table 2 to converge well in our experiments, LFP can also backpropagate rewards that do not have a corresponding loss function in the sense of Theorem 1, such as non-differentiable rewards, and still converge. To shortly demonstrate this, we repeat the experiment from Figure 3 on the first toy dataset ("Blob Dataset", cf. Appendix A.7.1) with the reward function from Equation 17.

As shown in Figure A.2, LFP is also able to converge in this setting under the right choice of hyperparameters (e.g., learning rate $3 \cdot 10^{-4}$). However, training is comparatively unstable and highly dependent on hyperparameters (cf. Figure 3, where training is far more stable and $100\%$ accuracy is reached for more than one learning rate under the same conditions using a different reward). As the reward from Equation 17 does not employ a convergence mechanism and leverages less fine-grained information of the closeness of the prediction to the ground truth (cf. reward reward #3 from Table 2), this is to be expected.

Nevertheless, this result shows that LFP converges in settings beyond those where it is equivalent to weight-scaled gradient descent (cf. Theorem 1).

### A.4  Differences between LFP and Gradient Descent Updates

Theorem 1 indicates equivalence between LFP and gradient descent under specific conditions, that are generally not met in practice (e.g., we utilize LFP-$\varepsilon$, which violates one of the conditions). Here, we show that if Theorem 1 is not met,

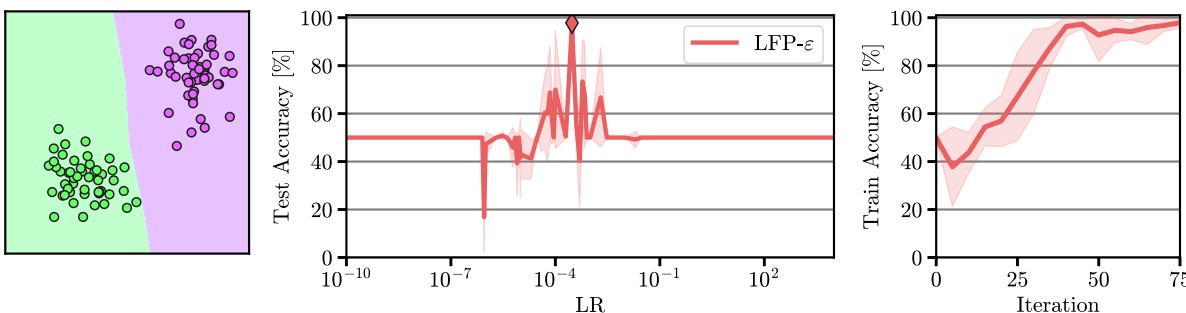

Figure A.2: LFP with reward #2 from Table 2 on toy data ("Blob Dataset"). *Middle*: Test accuracies over learning rates. The marker denotes the best-performing setting. *Left*: Decision boundary for the best-performing setting. *Right*: Train accuracy over training iterations for the best-performing setting. While training seems unstable compared to e.g. the results of Figure 3, there exists a choice of hyperparameters where LFP using this reward solves the task well. Note that the reward used here does not have a corresponding loss function in the context of Theorem 1. Results are averaged over 5 randomly drawn instances of the toy data, with the shaded area showing the standard deviation across draws.

LFP employs a fundamentally different update strategy, even on extremely simple models. For instance, the condition of specific piece-wise linear activations is not met by the SiLU non-linearity.

For this reason, we train a SiLU-activated one-neuron model for 10 iterations on a single datapoint using both LFP and gradient descent, with the respective criterion aimed at either maximizing or minimizing the neuron's output. This mimicks the optimization of a single neuron in a larger model, given all inputs and other parameters are kept constant. Weight initializations $W1 = (0.5, 0.5)$ and $W2 = (-1, -1)$ are deliberately placed to evaluate the pre-activation to the right and left of the SiLU minimum for the single used datapoint, $(1, 1)$. For maximization, we use an LFP-reward of $\text{sign}(o_c) \cdot (1 - \sigma(o_c))$ and gradient descent loss of $-\sigma(logits)$. For minimization, we use the respective negative criterion.

As shown by Figure A.3, weight updates of LFP do not follow the gradient when crossing into the decreasing area of the SiLU function (*top*). This is detrimental when optimizing for the minimum (*top left*), since the minimum is crossed as weights are constantly updated to decrease the pre-activation output. However SiLU has an upper bound of zero with decreasing pre-activations, and as a consequence, the resulting error missing the minimum is negligible (cf. the minor decrease in reward). I.e., in the worst case, SiLU will evaluate to zero and the neuron will simply not contribute to the output anymore. On the other hand, when optimizing for the (infinite) maximum (*top right*), this interaction of LFP with SiLU has a positive effect, as the model is updated in the correct direction of the *global* maximum (towards positive pre-activations), no matter the initialization. Interestingly, we observe a similarly ambivalent behavior for gradient descent. Here, updates are performed in the theoretically correct direction of the function slope, and the correct minimum is found (*bottom left*), however, in the maximization task, the success of gradient descent is initialization-dependent, with W2 becoming stuck in a local optimum (*bottom right*).

Consequently, both learning algorithms follow different update strategies with their respective strengths and weaknesses in their interaction with SiLU, with LFP finding the correct but unreachable maximum, while gradient descent converges to the correct minimum.

## A.5   A note on LRP-Rules

LFP utilizes LRP to propagate reward signals through the model. LRP introduces several specialized rules (Montavon et al., 2019; Achtibat et al., 2024) targeted at explaining specific types of layers or architectures, e.g., distinguishing between positive and negative contributions. One may thus ask whether these rules can be extended to LFP as well. The goals of LFP (training) and LRP (explaining) are not the same. Therefore, while some underlying paradigms of LRP-rules, such as numerical stability (cf. Section 3.1.3) align between both goals and the corresponding rules consequently translate well to LFP, several others do not. For instance, distinguishing between positively and negatively contributing connections would lead to updating those connections to different degrees, potentially destabilizing training and resulting in suboptimal solutions. We therefore do not extend most of these rules to LFP. Nevertheless, it may

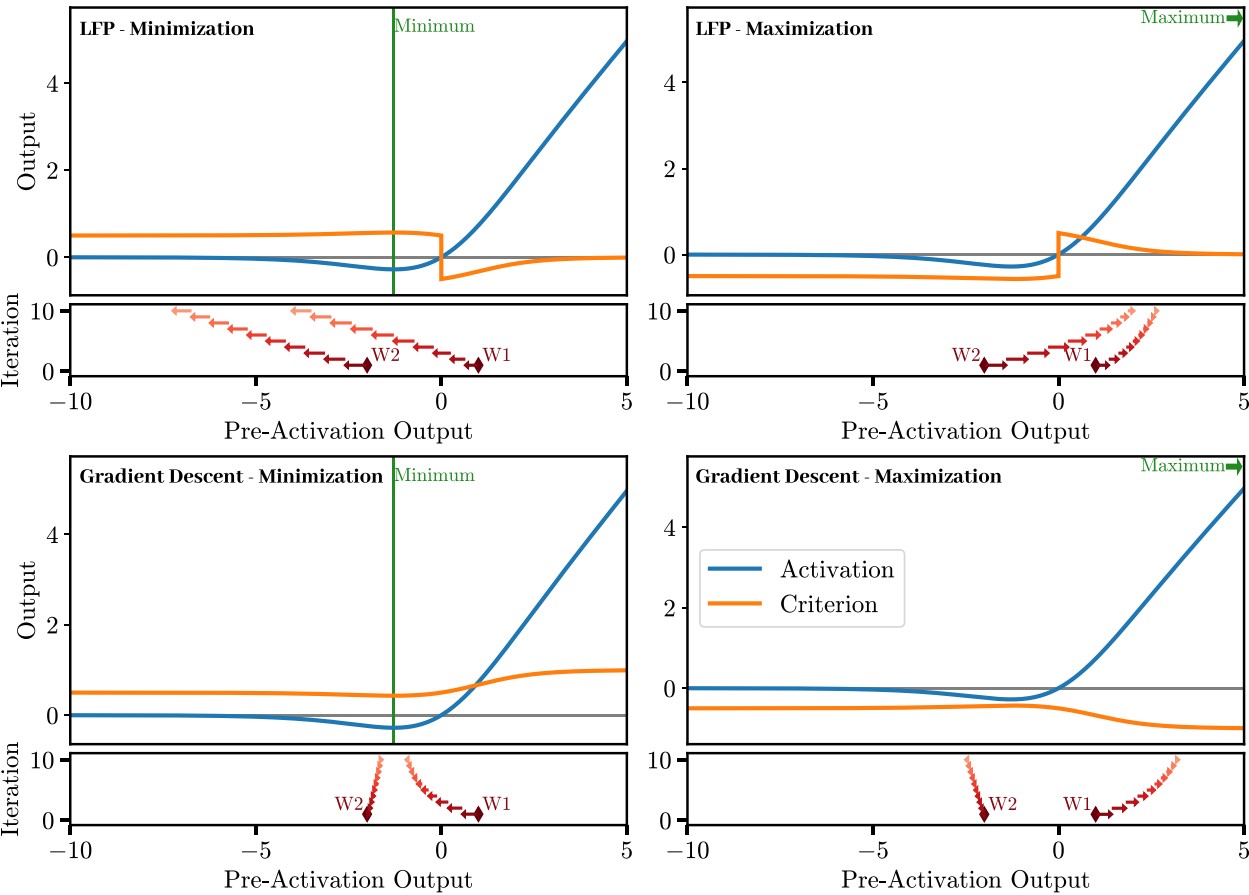

Figure A.3: Interaction of LFP (*top*) and gradient descent (*bottom*) with SiLU activation functions, shown on a toy model consisting of a single, SiLU-activated neuron trained on a single datapoint as input. The task is to minimize (*left*) or maximize (*right*) the neuron output. The respective optimum is shown in *green*, the post-activation output in *blue*, and the respective criterion (reward or loss) in *orange*. Starting from two different weight initializations (W1 and W2), model updates over 10 iterations are shown in *red*. For LFP, when crossing into the area left of the SiLU minimum, weights are updated in the (theoretically) wrong direction. We observe, however, that this behavior can be both detrimental (*top left*) and beneficial (*top right*) to finding the optimum. This non-monotoneous area can furthermore be detrimental to gradient descent as well, depending on the initialization (*bottom right*). We show in Table 3 that LFP is able to train SiLU-activated models without issue.

make sense (or even be necessary in order to apply LFP to state-of-the-art models) to develop completely new layer- or architecture-dependent rules for LFP (e.g., see Section 3.1.4, Appendix A.9, as well as Achtibat et al. (2024)), however, we leave further research into this direction to future work.

## A.6 Spiking Neural Networks

Spiking neural networks (SNNs) (Maass, 1997; Ponulak & Kasinski, 2011) are a special type of recurrent network built out of spiking neurons that model biological processing in real brains more closely than conventional neurons in DNNs. Spiking neurons have a membrane potential $U \in \mathbb{R}$ that dictates whether a neuron sends out a discrete spike or remains inactive. The output of neurons in a SNN can be modeled using the Heaviside step function:

$$H\left(U[t] - \theta\right) = \begin{cases} 1 & \text{if } U[t] > \theta \\ 0 & \text{else} \end{cases} \tag{A.2}$$

where

$$U[t] := \beta U[t-1] + W X[t] - \theta \, \mathrm{H}\left(U[t-1] - \theta\right). \tag{A.3}$$

In Equation A.3, the hyperparameter $\beta \in (0, 1]$ controls the decay rate of the membrane potential, while $\theta \in \mathbb{R}$ represents the threshold that determines the membrane potential required for the neuron to fire. The matrix $W$ represents the learnable parameters associated with the input connections from the preceding layer's output $X[t]$. The final term in Equation A.3 resets the membrane potential after the neuron fires. For further details, refer to Maass (1997) and Eshraghian et al. (2021).

## A.7 Details on Experiments

Of the experiments in this work, the ones in Sections 3.3, 3.4, and 3.6, as well as Appendix A.1, A.3, A.4, A.8, and A.9 ran on a *local* machine, while all other experiments ran on an *HPC Cluster*.

The local machine used Ubuntu 20.04.6 LTS, an NVIDIA TITAN RTX Graphics Card with 24GB of memory, an Intel Xeon CPU E5-2687W V4 with 3.00GHz, and 32GB of RAM.

The HPC-Cluster used Ubuntu 18.04.6 LTS, an NVIDIA A100 Graphics Card with 40GB of memory, an Intel Xeon Gold 6150 CPU with 2.70GHz, and 512GB of RAM. Apptainer was used to containerize experiments on the cluster.

The code for all experiments was implemented in python, using PyTorch (Paszke et al., 2019) for deep learning, including weights available from torchvision, as well as snnTorch (Eshraghian et al., 2021) for SNN applications. matplotlib was used for plotting. The implementation of LFP builds upon zennit (Anders et al., 2021) and LXT (Achtibat et al., 2024), two XAI-libraries.

Seeds were chosen using the *$RANDOM* shell command, and the following function was used to set them for the experiments:

```python
import torch
import numpy as np
import os
import random

def set_random_seeds(seed):
    torch.manual_seed(seed)
    torch.cuda.manual_seed(seed)
    torch.cuda.manual_seed_all(seed)
    np.random.seed(seed)
    random.seed(seed)
    torch.manual_seed(seed)
    os.environ['PYTHONHASHSEED']=str(seed)

    torch.backends.cudnn.benchmark=False
    torch.backends.cudnn.deterministic=True
    torch.backends.cudnn.enabled=False
```

### A.7.1 Proof-of-Concept

Small ReLU-activated MLPs were trained on three toy datasets with LFP-$\varepsilon$ and reward #3 from Table 2 (as well as stochastic gradient descent with categorical cross-entropy loss) using a batch-size of 128 and momentum of 0.95. The MLPs each consist of three dense layers with 32, 16, and $n$ neurons, respectively, and are visualized to the *right* of Figure 3. $n$ refers to the number of classes, which vary between 2 and 3 depending on the dataset. The models were trained for a large number of different learning rates, chosen according to the following formula: $a * 10^b$, with $a \in [1, 2, ..., 10]$ and $b \in [-10, -9, ..., 3]$. We used the following toy datasets:

*Blob Dataset*: 1000 training samples and 100 test samples were drawn using the scikit-learn function `skdata.make_blobs` with two classes, centered at $[1, 1]$ and $[2, 2]$, respectively, and a cluster standard deviation of 0.2. Models were trained on this dataset for 10 epochs.

*Circle Dataset*: 10000 training samples and 500 test samples were drawn using the scikit-learn function `skdata.make_circles` with two classes, using a cluster standard deviation of 0.2 and a scale factor of 0.05. Models were trained on this dataset for 10 epochs.

*Swirl Dataset*: 10000 training samples and 500 test samples were drawn for three classes in the shape of a swirl, similar to the toy dataset used in Yeom et al. (2021). Models were trained on this dataset for 15 epochs.

Refer to the supplied code[11] for more specifics on dataset generation. We did not employ seeding for this experiment but instead averaged all results over five different versions of each dataset.

For the results shown in Figure 4, we instead used a version of the *Blob Dataset* with classes centered at $[-1, -1]$ and $[1, 1]$, respectively. We further employed a single-neuron model without bias trained using either gradient descent and binary cross-entropy loss or LFP-$\varepsilon$ and reward #4 from Table 2. We trained this model for one epoch using a learning rate of $0.5$, batch size of $8$, and no momentum. Initial weights were set manually at $[0, 4]$, $[4, 0]$, $[2, 4]$, $[4, 2]$, $[-4, -4]$, $[-4, 4]$, and $[4, -4]$ to cover some different proportions of $w_1$ to $w_2$. Results are shown for one seed only.

### A.7.2 Computational Cost of LFP

We trained a small three-layer MLP and LeNet for 50 epochs on the MNIST dataset. The MLP consists of 120, 84 and 10 neurons, respectively. The LeNet utilizes two convolutional layers with 16 channels and kernel size of 5, followed by two maxpooling layers, respectively. For classification, three linear layers (120, 84, 10 neurons) with two 50% dropout layers in between are used. Both models utilize ReLU activations, except for the last layer.

These models are trained using the following methods and hyperparameters:

**LFP-$\varepsilon$.** We utilize a batch-size of 128, learning rates $0.1$ and $0.01$ for the MLP and LeNet, respectively, a momentum of $0.9$, an SGD optimizer and the reward #3 from Table 2.

**Gradient Descent.** We utilize a batch-size of 128, learning rates $0.1$ and $0.01$ for the MLP and LeNet, respectively, a momentum of $0.9$, an SGD optimizer and a categorical crossentropy loss.

**Feedback Alignment.** We adapt an implementation[12] of the original method Lillicrap et al. (2016) and utilize a batch-size of 128, learning rates $0.1$ and $0.01$ for the MLP and LeNet, respectively, a momentum of $0.9$, an SGD optimizer and a categorical crossentropy loss.

**PSO.** We adapt an implementation (Sansom, 2022)[13] of the original method (Eberhart & Kennedy, 1995) and utilize a batch-size of 1000, cognitive and social coefficients of 2, an inertial weight of $0.8$, 1000 particles, a search space of $[-0.1, 0.1]$ and categorical crossentropy loss as a measure of fitness. To encourage exploration, we decay cognitive and social coefficients in each particle by a factor of $0.95$ for every iteration where the particle or swarm optimum, respectively, was not updated. Coefficients are reset to the initial value when an update occurs, to encourage search near the new optimum.

**dlADMM.** We utilize the implementation[14] from the original paper Wang et al. (2019), only adapting the number of neurons in the MLP to match the MLP used in the other methods. An implementation compatible with the LeNet was not available.

### A.7.3 Non-ReLU Activations

For this experiment, we applied LFP-$\varepsilon$ with reward #3 from Table 2 and stochastic gradient descent with categorical crossentropy loss to three different model architectures: A randomly initialized "VGG-like" model, consisting of three convolutional blocks with two kernel-size 3 and stride 1 convolutional layers and a kernel-size 2 and stride 2 max-pooling layer each, followed by three fully-connected layers with two 50% dropout layers in-between. In order, the convolutional layers have the following number of channels: 32, 64, 128, 128, 256, 256. The linear layers have the following number of neurons: 1024, 512, *number of classes*. We further used the VGG-16 (Simonyan & Zisserman, 2014) and ResNet-18 (He et al., 2016) models, initialized by ImageNet weights available in *torchvision* (Paszke et al., 2019).

Out of the above models, we trained the VGG-like model on the CIFAR10 and CIFAR100 tasks (Krizhevsky, 2009) for 50 epochs with a batch size of 128 using a one-cycle learning rate schedule (Smith & Topin, 2017) with a max learning

---

[11] https://github.com/leanderweber/layerwise-feedback-propagation
[12] https://github.com/jordan-g/PyTorch-Feedback-Alignment-Layers
[13] https://github.com/qthequartermasterman/torch_pso
[14] https://github.com/xianggebenben/dlADMM

rate of 0.1, and gradient (in this case reward) norm clipping with a max norm of 3. The VGG-16 and ResNet-18 models were trained on the Caltech-UCSD Birds-200-2011 (CUB) (Wah et al., 2011), ISIC 2019 (Tschandl et al., 2018; Codella et al., 2018; Combalia et al., 2019) skin lesion, and Food-11[15] classification datasets. Here, models were trained for 100 epochs with a batch size of 32, employing the one-cycle learning rate schedule (Smith & Topin, 2019) with a max learning rate of $1e^{-3}$ (except for VGG-16 on Food-11, which performed better with a max learning rate of $1e^{-4}$), and a weight decay of $1e^{-4}$.

All models were trained multiple times, with ReLU, SiLU (Elfwing et al., 2018), ELU (Clevert et al., 2016), Tanh, Sigmoid, and Heaviside step functions for hidden layer activations. Results are averaged over three randomly generated seeds.

For more details regarding data and training hyperparameters, please refer to the supplied code.

### A.7.4 Application 1: Sparse and Prunable Models

For this experiment, we utilized the same datasets as detailed in Appendix A.7.3: Non-ReLU Activations. In addition to the models used in Appendix A.7.3: Non-ReLU Activations (only the ReLU-activated versions), we also employed ResNet-32 (He et al., 2016) and VGG-16 with BatchNorm (VGG-16-BN) (Simonyan & Zisserman, 2014) for the CUB, ISIC2019, and Food-11 datasets. To these, we applied LFP-$\varepsilon$ with reward #3 from Table 2 and stochastic gradient descent with categorical crossentropy loss. We further used the same training setups as in Appendix A.7.3: Non-ReLU Activations, except for slightly different learning rates: On both CIFAR10 and CIFAR100, we trained with a max learning rate of 0.1. For VGG-16 and VGG-16-BN, we used a max learning rate of $1e^{-3}$ and for ResNet-18 and ResNet-32, a max learning rate of $5e^{-3}$.

For visualizing the weight distributions in Figures 7, A.6, A.7, and A.8, we first apply BatchNorm canonization as described below, and then divide the unsigned weights by their maximum value.

For pruning, we only consider methods without subsequent re-training or fine-tuning of the model and always prune a given percentage of connections (as opposed to, e.g., pruning all connections with some importance score below a given threshold). We only consider *unstructured* pruning, and only apply pruning to convolutional layers, as done by Yeom et al. (2021). We also *canonize* all BatchNorm layers into the preceding convolutional layers before computing pruning criteria (resulting in a functionally equivalent model), for two reasons: firstly, to be able to correctly apply the relevance-based pruning criterion (Yeom et al., 2021), and secondly, to be able to apply the global magnitude-based pruning criterion, since BatchNorm layers can lead to arbitrary weight magnitudes in the preceding layer as they are invariant to weight scale (van Laarhoven, 2017). To determine the pruning criterion, we applied the following algorithms:

*Magnitude-based Pruning*: Here, connections are simply pruned based on the magnitude of their weights, similar to Han et al. (2015). It is assumed that the lowest magnitude weights are the least important, and consequently, they are removed first. We consider both *local* and *global* variants of magnitude-based pruning. The former applies the pruning criterion layer-wise, removing the same percentage of connections from each layer at each pruning step. The latter considers all layers at the same time when computing the pruning criterion, removing a percentage of connections from the whole model at each pruning step. *Global* pruning usually performs better but can lead to layer collapse (Tanaka et al., 2020).

*Relevance-based Pruning*: As suggested by Yeom et al. (2021), we also apply a *global* relevance-based pruning criterion. Specifically, we apply LRP-$\varepsilon$ to the model (this is different to Yeom et al. (2021), who use LRP-$z^+$), summing relevances of each connection across the whole test dataset. The absolute summed relevance of each connection is used as a criterion for pruning.

All results in this experiment are averaged over three randomly generated seeds. For more details regarding data, training, and pruning hyperparameters, please refer to the supplied code.

### A.7.5 Application 2: Spiking Neural Networks

For the results in Figure 9, LeNet and multilayer perceptron (MLP) architectures were trained on the MNIST dataset (LeCun et al., 1998) for image classification. The LeNet architecture comprised two successive convolution blocks

---

[15]https://www.kaggle.com/datasets/vermaavi/food11

followed by a final MLP block. Each convolution block contained a convolution layer (12 and 64 channels, respectively) with 5×5 kernels, stride 1, and no input padding. These were followed by max-pooling layers (2×2 kernels, stride 2) and Leaky Integrate-and-Fire (LIF) activation functions (described in Equation A.2-Equation A.3). The final MLP classification block consisted of a dense layer with 1,024 LIF neurons, followed by another LIF non-linearity.

The MLP architecture consisted of three dense layers each followed by the LIF non-linearity. The dimension of the hidden layers was set to 1,000.

For the LIFs activation function, we adapted the implementation from Eshraghian et al. (2021). To enable gradient-based training, we employed a surrogate function $s\colon \mathbb{R} \to \mathbb{R}$ in the backward pass, defined as:

$$s(x) = \frac{x}{1 + 25 \cdot |x|}$$

We furthermore used a fixed LIF threshold of $\theta=1$ and varying the decay factor $\beta$ to values of 0.3, 0.6, and 0.9. The sequence length $L$ was varied across 5, 15, and 25 time steps. Both networks were trained for three epochs using a batch size of 128. For optimization, we used the stochastic gradient descent algorithm. Learning rates of $10^{-3}$, $10^{-2}$, $5 \cdot 10^{-2}$ $7.5 \cdot 10^{-2}$, 0.1, 0.25, 0.5 and 0.8 were investigated in combination with the one-cycle-lr-scheduler proposed in Smith & Topin (2019). For stability, in this experiment, the backward pass was normalized by the maximum absolute value between layers. Additionally, we utilized the *adaptive gradient clipping* strategy proposed in Brock et al. (2021). The experiments were performed across three different random seeds, with results averaged.

For the gradient-based training, the categorical cross-entropy loss was employed between the one-hot encoded target classes and the output spikes per step. The resulting values were averaged across the batch and the number of steps to obtain the final loss[16].

The LFP training methodology requires a reward function. For the SNNs experiments, the static MNIST images are encoded into the required input sequence through constant encoding. Specifically, a given image $X_i \in \mathbb{R}^{w \times h \times c}$ is fed sequentially into the network $n$ times, yielding the input $X \in \mathbb{R}^{n \times w \times h \times c}$. When presented with the input $X$, the SNNs produces an output sequence $(o_{k,i})_{i=1}^{n}$ per class, where $1 \leqslant k \leqslant m$ is the class index. This can be written as an output matrix $O \in \mathbb{R}^{n,m}$. The predicted class of the model is determined by accumulating the spikes for each class and taking the maximum. For a given model output with corresponding target label $y_c \in \mathbb{N}$, we rewarded the network with the matrix $R \in \mathbb{R}^{n,m}$, elementwise defined via

$$r_{c,i} = \begin{cases} 1 - \sigma(\sum_i o_{c,i} - \frac{n}{2}) & \text{if } y_c = c \\ \sigma(\sum_i |o_{c,i} - 1| - \frac{n}{2}) - 1 & \text{if } y_c \neq c \end{cases} \tag{A.4}$$

where $\sigma$ denotes the sigmoid function. This reward function encourages spikes for the correct class while discouraging spikes for others, yet tolerates some spikes across all classes to yield a more stable signal that is less affected by noise.

For the experiments in Figures 10 and 11, we utilized the implementation of Mozafari et al. (2018) in `www.github.com/aidinattar/snn` for the R-STDP training. Here, the Cifar10 data was encoded as previously outlined for MNIST. We implemented two convolutional neural networks, utilizing 25 time-steps per sample and $\beta = 0.9$, adapting the `deeper2024` and `resnet2024` models from the R-STDP codebase. In conducting gradient-based and LFP training, we modified the architectures by adding a final fully connected layer to facilitate prediction. In our revised architectures, each block is composed of a convolutional layer, succeeded by a max pooling layer and a LIF-layer. R-STDP ran only on the (original) `deeper2024` model from the R-STDP codebase.

To enable gradient-based training, we employed a shifted arctan surrogate function in the backward pass, as per (Fang et al., 2021). We investigated learning rates of $1e^{-3}$, $5e^{-4}$, and $1e^{-4}$ for gradient descent and LFP, where the setting where the maximum accuracy was reached (lr $5e^{-4}$) are reported in the figure. For R-STDP we used the hyperparameters of the reference implementation, only changing the number of time-steps to 25. We trained the LFP and gradient descent models for 50 epochs each. For R-STDP, each layer was separately trained for 10 epochs. Refer to the implementation[17] for further details.

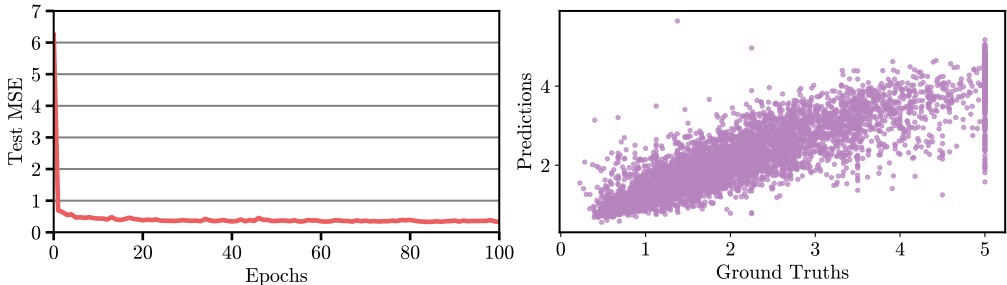

Figure A.4: LFP-regression on the California Housing dataset. We report the MSE on the test set *(left)* and a scatter plot of (test set) prediction vs. ground truth house prices *(right)*. Note that house price 5 is equally distributed across variation of the most predictive features in the dataset (despite strong correlation with house price and e.g. median income overall) causing the random behavior of predictions at ground truth 5.

## A.8 Extending LFP to Regression

In this work, we have focused on applying LFP for classification. In the following, we shortly outline how LFP can be extended to regression models. This requires two adaptations from the classification-formulation in the paper: (1) A different reward function and (2) (possibly) a reference point correction in the last, linearly activated layer.

The former (1) is relatively trivial. Given some examples $(x, y)$ of the ground truth function $y = g(x)$, and the model output $o_c$, suitable initial rewards $o_c$ include, e.g., $o_c = (y_c - o_c) \cdot \text{sign}(o_c)$ and $r_c = (y_c - o_c)^3 \cdot \text{sign}(o_k)$. Note that $c$ does not correspond to a class, but rather an output dimension of the regression model here. These rewards consider the direction of the difference between ground truth and prediction, and are proportional to rewards derived from gradient descent with MSE and MSE-squared in the sense of Theorem 1, respectively.

The second (2) adaptation is more complex. LFP in the present classification setting implicitly utilizes a reference point of zero when evaluating contributions, measuring how a $z_{ij}^{[l+1]}$ shifts $z_j^{[l+1]}$ away from zero (the reference point). For the last, linearly activated layer of regression models, zero may not always be an ideal choice, and may require a dedicated rule for successful application in regression scenarios. For applications of LRP in regression settings, the choice of reference activation value is well motivated and explained in Letzgus et al. (2022). Choosing a suitable reference value for LRP depends on the intended explanation target. However, setting such reference during active training, might be non-trivial, as it strongly affects the magnitude and direction of parameter updates, and should be investigated in dedicated future work.

We provide an initial experiment training a small MLP regressor for 100 epochs on the California Housing Dataset with LFP (predicting house prices). The MLP consists of three layers with 256, 128, and 1 neurons, respectively. Except for the last, linearly activated layer, we use ReLU activations, and apply dropout with a probability of $50\%$ after the first two linear layers during training. We utilize a batch size of 128, a learning rate of 0.05, a momentum of 0.9, and $r_k = (y_k - o_k)^3 \cdot \text{sign}(o_k)$ and do not change the reference value from zero. In this example, the application of LFP can be considered successful, achieving a final MSE of 0.33 on the test set. Notably, in this simple application, adaptation of the reference value does not seem necessary.

## A.9 Extending LFP to Transformer Models

While we do not investigate this in-depth in this work, the following Section briefly discusses how LFP could be extended to train state-of-the-art transformer architectures.

These models mainly employ GeLU, SiLU or ReLU activations, which are handled by LFP out of the box, as they are sign-preserving (e.g., cf. Table 3 for ReLU and SiLU; GeLU has a similar shape as SilU). They further employ LayerNorm (Ba et al., 2016) instead of BatchNorm. LayerNorm normalizes across features instead of the batch, and can thus be handled using the same rule as we used for BatchNorm (equation 11 and equation 12), except with mean and variance computed across features instead of samples.

---

[16]This corresponds to the *"cross entropy spike rate loss"* in the implementation of Eshraghian et al. (2021).
[17]https://github.com/leanderweber/layerwise-feedback-propagation

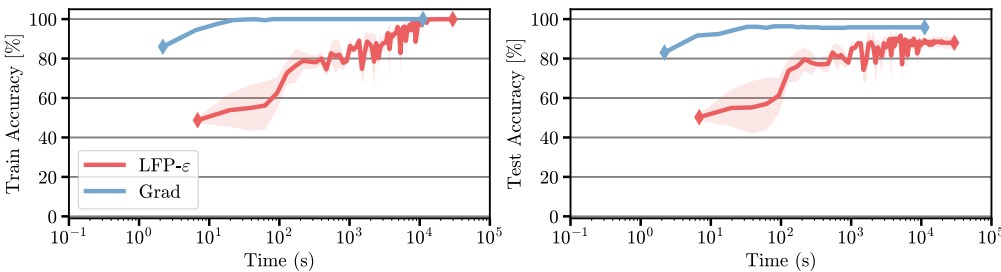

Figure A.5: LFP performance on a ViT over clock time (s). We report training *(left)* and test *(right)* accuracy for gradient descent and LFP. While LFP provides a valid training signal even in such deep architectures, it is outperformed by gradient descent, emphasizing the need for further research into more optimal propagation rules. Results are averaged over 3 seeds, with standard deviation highlighted around each line.

In transformer models, the main challenge for LFP is propagation through attention layers, specifically the highly nonlinear softmax function in the attention. The issue is described and solved in Achtibat et al. (2024) for LRP, but obtaining neuron contribution (which LFP relies on) specifically for attention layers is still an active field of research.

The below experiment shows LFP applied to a Visual Transformer (Dosovitskiy et al., 2021), handling GeLU and LayerNorm as described above, and otherwise adapting the rules suggested by Achtibat et al. (2024). These can be directly translated to LFP since they do not alter propagation through any parameterized operations. We utilize a model pretrained[18] on ImageNet and fine-tune it on a small bean disease classification dataset[19] for 100 epochs, altering all parameters except for the embeddings. We utilize a batch-size of 32, momentum of 0.9 and learning rate of $2e^{-4}$ with an Adam (Kingma & Ba, 2015) optimizer. While LFP provides a meaningful training signal, with only a slight increase in training time per epoch, it is outperformed by gradient descent in this experiment and converges much slower, emphasizing the need for further research to extend it to state-of-the-art model architectures.

## A.10 Additional Figures

---

[18]https://huggingface.co/google/vit-base-patch16-224-in21k
[19]https://huggingface.co/datasets/AI-Lab-Makerere/beans

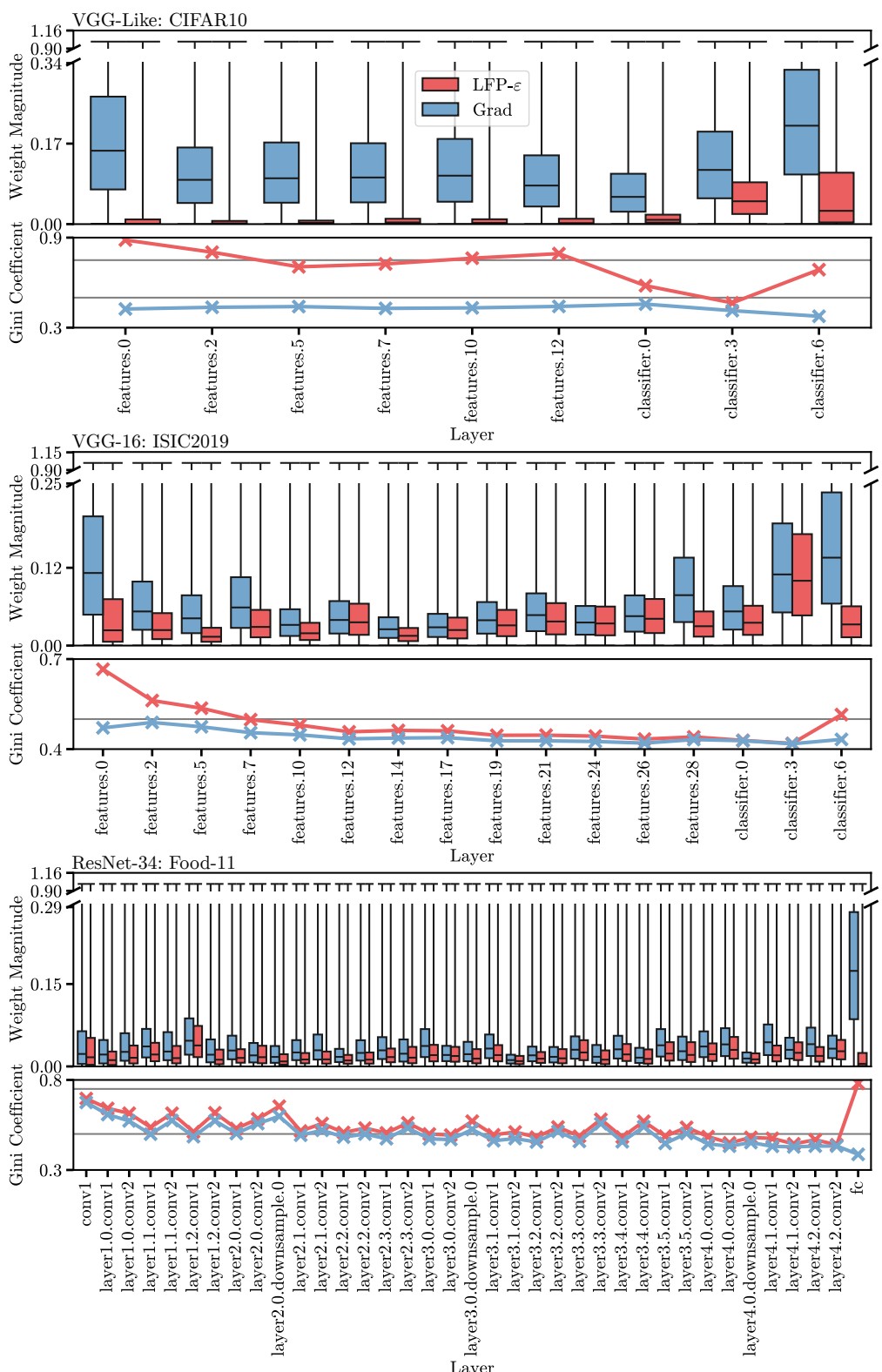

Figure A.6: More normalized unsigned weight distributions, confirming observations on Figure 7.

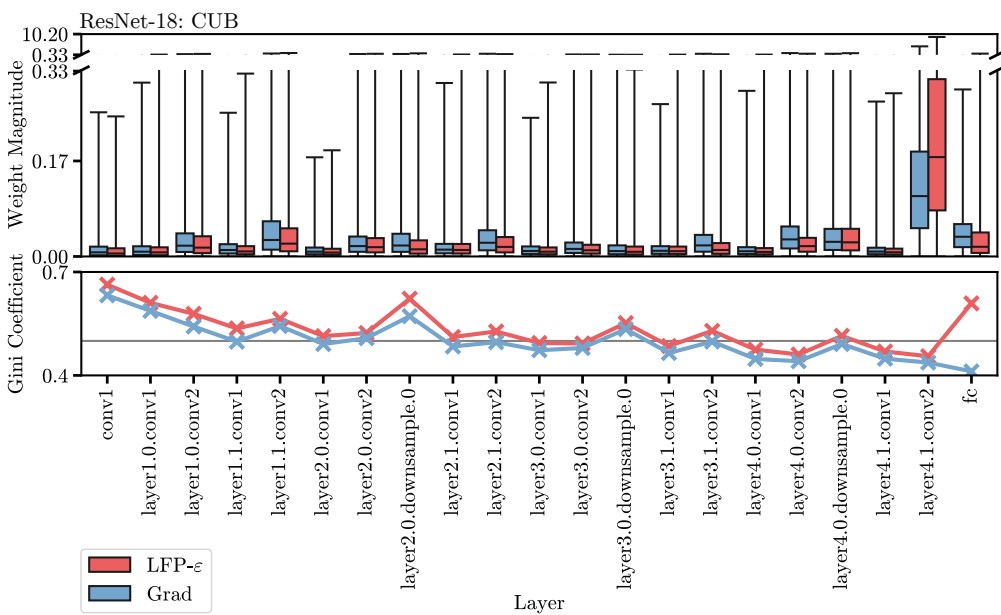

Figure A.7: Original unsigned weight distributions, without maximum value normalization. (cf. Figure 7).

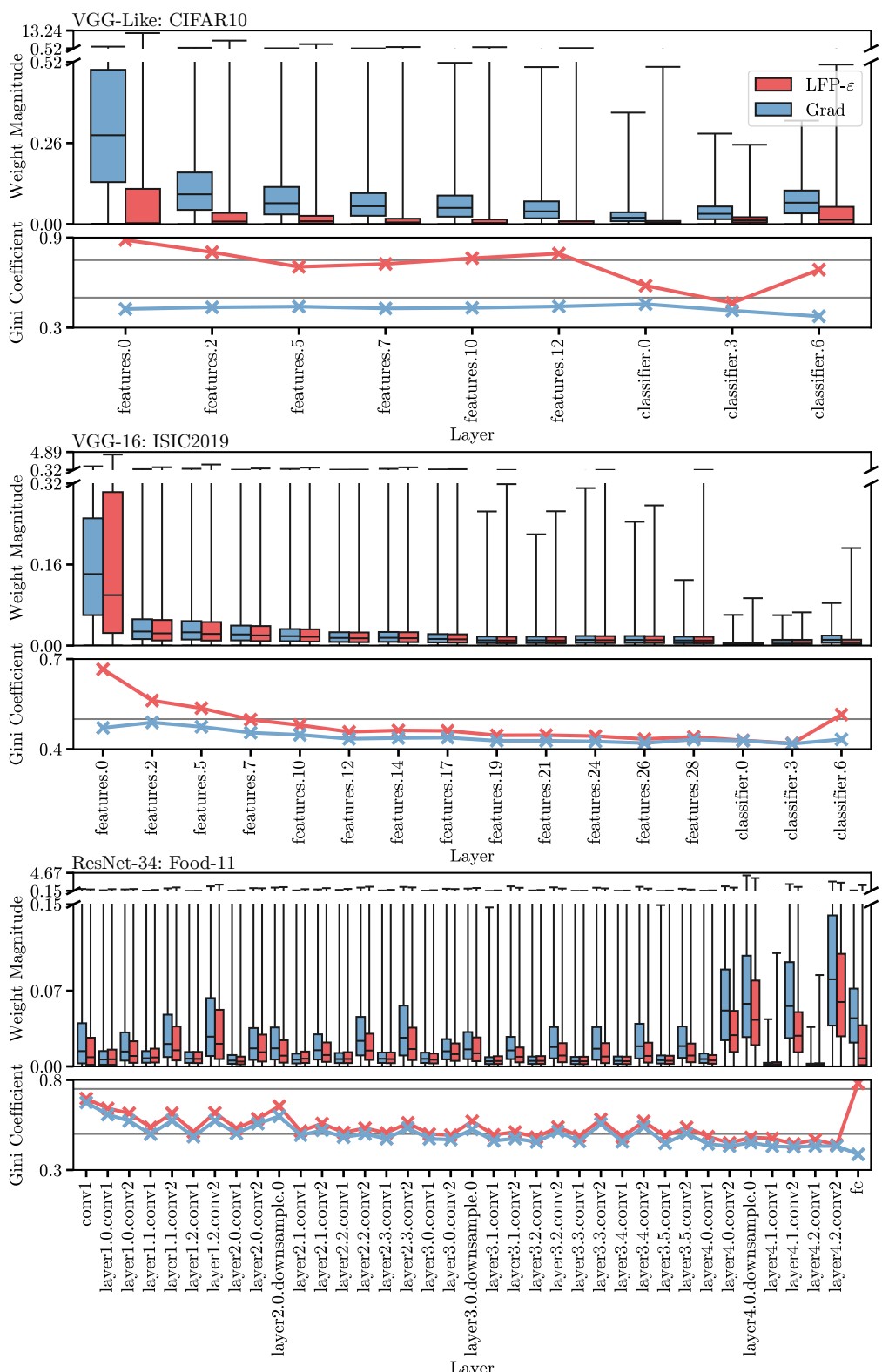

Figure A.8: Original unsigned weight distributions, without maximum value normalization. (cf. Figure A.6).

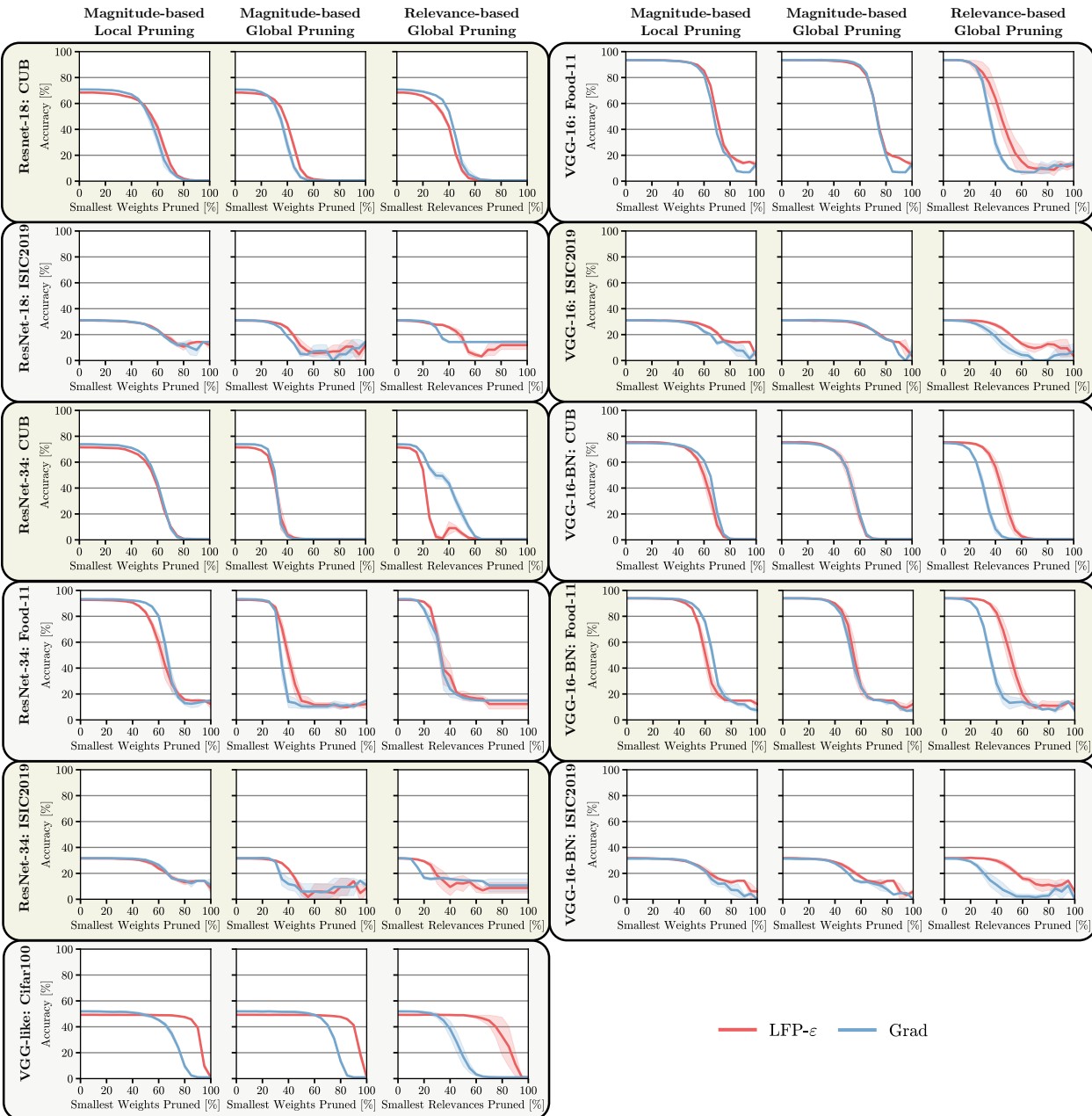

Figure A.9: Pruning results, extending Figure 8. Note that we report the weighted accuracy for ISIC2019, due to the severe imbalance of that dataset. While there is slight variance depending on the setting, and there are specific cases where gradient-trained models seem more prunable at least for one out of the three criteria, the general trend from Figure 8 holds, with LFP-trained models retaining accuracy for longer in the majority of cases.

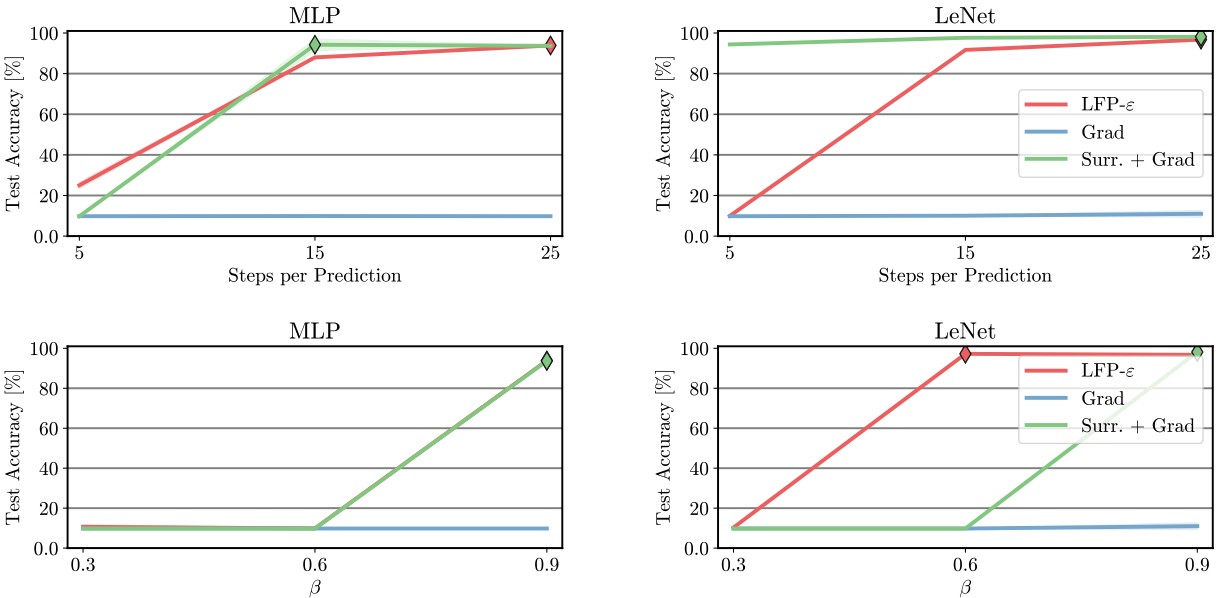

Figure A.10: Final test set accuracies for different hyperparameter settings in MLP and LeNet based SNNs. *Top row*: Variation with sequence length of the encoded data. Values used: 5, 15, and 25 steps. *Bottom row*: Variation with decay factor $\beta$. Values used: 0.3, 0.6, and 0.9. Three training methods (LFP-$\varepsilon$, gradient descent (Grad), and surrogate-enabled gradient descent (Surr. + Grad)) are compared in each graph.

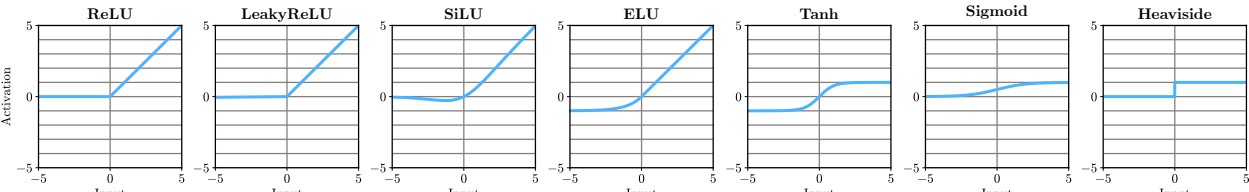

Figure A.11: Illustration of the activation functions used in Section 3.5 and throughout the paper.

