# OpenReview forum: "Efficient and Flexible Neural Network Training through Layer-wise Feedback Propagation"
_TMLR — Accepted by TMLR_

### Review · Reviewer_LAuB · 2025-03-06

**Summary Of Contributions:**

The paper proposes a new method for training neural networks that does not require gradients (though it still uses a backward pass).

The method relies on a concept of "reward", quite distinct from how this term is generally used. For a given neuron, this "reward" specifies whether we want the neuron to amplify its current firing (negative or positive), or whether we want it to do the opposite (fire less if its firing was positive, fire more if it was negative).

The authors propose rules which can backpropagate this "reward" through a multi-layer architecture (i.e. compute the "reward" for layer l-1 from rewards for layer l), under certain conditions on the activation function.

They also propose a weight update rule based on this reward and the activations of the neurons (and the current weights). Somehow this rule happens to be a "reward"-modulated Hebbian rule, but where the post-synaptic output *divides* the pre-synaptic input, instead of multiplying it.

Under some conditions, and with a suitably chosen top-level "reward", this rule is equivalent to gradient descent.

Various experiments show that this method can train deep neural networks, again under some conditions on the activation function. It can handle non-differentiable objectives and activation functions that would make  gradient descent unusable, though also introduces new challenges (e.g. no updates if activation is zero).

**Audience:**

Yes

**Broader Impact Concerns:**

No broader impact concerns.

**Claims And Evidence:**

Yes

**Requested Changes:**

Overall the paper is clear and self-contained.

IIUC your update rule looks like a "reward"-modulated learning rule, except that now the post-synaptic output divides the pre-synaptic input, instead of multiplying it. Please elaborate a bit on that. The connection, and the difference, are too glaring to not be mentioned.

p. 7: "Equations equation"

**Strengths And Weaknesses:**

Strength:

- The method seems to be able to train neural networks without using gradients, under some conditions.

- The method seems to work.

- Importantly, while this method is highly restricted in terms of activation functions and outcomes, these limitations are clearly and honestly described in detail.

Weaknesses:

- The restrictions on the method are indeed quite strong. In particular, there are strong constraints on allowable activation functions (am I correct that they should always preserve the sign of the incoming net input?) and the "reward" function must be carefully chosen to produce adequate results.

- The method can result in "dead" neurons that stop updating, though IIUC the reason is quite different than in traditional SGD training  (zero output rather than zero gradient).

---

> ### Author Response · Authors · 2025-03-21
> **Some notes on the connection to Hebbian Learning and Weaknesses of LFP**
>
> Thank you for the feedback, especially for pointing out the connection to Hebbian Learning.
>
> ### Relation to Hebbian Learning
> While we derived equations (2-4) from the relevance propagation of LRP, as described in (Bach, 2015), there is indeed a close similarity of the resulting update rule to Hebbian learning. I.e., with some approximations, we can reformulate the LFP-Update Rule as
>
> $$d_{w_{ij}}^\text{lfp} = \frac{|w_{ij}| \cdot a_i}{z_i}\cdot r_j \propto \frac{a_i}{z_j} \cdot r_j  \approx \frac{a_i}{a_j} \cdot r_j$$
>
> Which (excluding zeros) has the same sign as a hebbian update $d_{w_{ij}}^{\text{hebb}} = a_i a_j$, but modulated by a reward $r_j$. Where $d_{w_{ij}}^{\text{hebb}}$ simply results in increasing $|a_j|$, LFP contextualizes $a_j$ to a given task by multiplying with $r_j$, decreasing $|a_j|$ if neuron $j$ hinders performance as evaluated by the reward function.
> Aside from this reward modulation, as you pointed out, there is also the division of $a_i$ and $a_j$ in place of multiplication. While this preserves the update direction of $d_{w_{ij}}^{\text{hebb}}$, it also implicitly regularizes the update when the weights grow too large (and, as a consequence, also $|a_j|$), similar to, e.g., Oja's rule.
> Lastly, the LFP-update also includes a multiplication with the weight magnitude that induces sparse updates, which have been shown to improve performance specifically in the context of Hebbian learning (Gupta, 2021).
> We will include this in the paper.
>
> ### Activation Function Constraints
> Regarding the activation functions, your understanding is correct in that they should preserve the sign of the incoming net input. This constraint aligns well with most practically applied activations (ReLU, Heavyside, Tanh, …) but is quite restrictive considering all possible activations. The reason for this constraint is an underlying assumption of sign preservation in using the $r_i$ of Equation 3 as the $r_j$ of Equation 4. If the pre- and post-activation signs differ, one could circumvent this constraint by setting $r_j \text{(Eq. 4)} = -r_i \text{(Eq. 3)}$  in the backward propagation.
>
> ### Dead Neurons
> Regarding the dead neurons, again, your understanding is correct that they can theoretically occur due to zero outputs $a_i$, which will block any backwards reward propagation at that neuron. This is due to LRP, which is used for reward propagation, assuming zero to be a neural value with no contribution. In practice, we found that dead neurons were not hindering convergence, depending on network and optimization, as using a momentum and the time dynamics of SNNs seem to mitigate neurons dying.
>
> ### Citations
>
> *Sebastian Bach, Alexander Binder, Grégoire Montavon, Frederick Klauschen, Klaus-Robert Müller, and Wojciech Samek. On pixel-wise explanations for non-linear classifier decisions by layer-wise relevance propagation. PLoS ONE, 10(7):e0130140, 2015.*
>
> *Manas Gupta, Arulmurugan Ambikapathi, and Savitha Ramasamy. Hebbnet: A simplified hebbian learning framework to do biologically plausible learning. In IEEE International Conference on Acoustics, Speech and Signal Processing, ICASSP 2021, Toronto, ON, Canada, June 6-11, 2021, pp. 3115–3119. IEEE, 2021.*

---

> ### Author Response · Authors · 2025-04-05
> **Included Related Work and Section on Hebbian Learning**
>
> We have updated the paper with additional related work on Hebbian learning, and included an additional Section (newly added 3.1.2)  discussing the relationship between LFP and Hebbian learning.

---

### Review · Reviewer_t9tc · 2025-03-21

**Summary Of Contributions:**

The authors propose a training methodology called Layer-wise Feedback Propagation (LFP), which trains the underlying network by distributing rewards to individual neurons based on their contributions. This approach enables a greedy optimization strategy, strengthening or weakening connections based on their contributions (without using differentiation operation), to enhance learning efficiency. The authors validate the proposed approach through a combination of theoretical analysis and empirical studies, demonstrating its effectiveness.

**Audience:**

Yes

**Broader Impact Concerns:**

There are no strong concerns on the ethical implications of the work.

**Claims And Evidence:**

Yes

**Requested Changes:**

(a) Additional literature review and comparison with other gradient free training approaches such as equilibrium propagation.
(b) Clock time comparison in experiments with other training approaches. This is to understand any computational overhead associated with training with this approach.
(c) If possible please add some additional experiments on SNNs.

Questions:
(a) How will the authors approach training regression based models with this technique?
(b) How to tackle activations like gelu, which are widely used in transformer-based architectures. Also, how to incorporate Layernorm?

**Strengths And Weaknesses:**

Strengths:
(a) The authors present an interesting perspective on differentiation-free model training, highlighting the concept of reward-based learning at the neuronal level.
(b) The methods discussed is highly relevant for on-chip learning scenarios.

Weaknesses:
(a) The performance of this approach is not comparable to current sota, even considering gradient free approaches like equilibrium propagation. Thus scalability is a concern.
(b) To further strengthen the case for the proposed method, additional experiments on SNNs could have been conducted, comparing the approach against training paradigms such as STDP, Decolle, BPTT with surrogate gradients, etc.

---

> ### Author Response · Authors · 2025-04-03
> **Response w.r.t. Scalability, Regression and Transformers**
>
> Thank you for your feedback! As you suggested, we will include a review of and comparison with additional gradient-free and/or biologically (more) plausible approaches such as Equilibrium Propagation, Decolle, STDP, Target Propagation, Feedback Alignment, among others in a later version this week.
>
> ### Scalability of LFP
> Thank you for pointing this out. We have included a benchmarking experiment in the (new) Section 3.4 for a small MLP and LeNet on MNIST, to compare clock time of LFP against BP, Feedback Alignment, dlADMM (if applicable), and Particle Swarm Optimization on ANN. Note that we do not compare LFP against Equilibrium Propagation empirically, as adapting our implementation to convergent RNNs would require significant alterations and additional time; however, we will include EP and similar approaches in the Related Work. As shown by the (new) Figure 5, LFP requires computation times similar to gradient descent and feedback alignment.
>
> We demonstrate the scalability of our approach to large architectures in Table 3 and in applying LFP to transformer architectures (see below).
>
> ### Application to Regression Models
> Extending LFP to regression models requires two adaptations from the classification-formulation in the paper. (1) A different reward function and (2) (possibly) a reference point correction in the last, linearly activated layer.
>
> This is relatively trivial. Given some examples $(x, y)$ of the ground truth function $y = g(x)$, and the model output $o_k$, suitable initial rewards $r_k$ include, e.g., $r_k = (y_k-o_k) \cdot \text{sign}(o_k)$ and $r_k = (y_k-o_k)^3 \cdot \text{sign}(o_k)$. These rewards consider the direction of the difference between ground truth and prediction.
> LFP in the present classification setting implicitly utilizes a reference point of zero when evaluating contributions, measuring how a $z_ij$ shifts $z_j$ away from zero (the reference point). For the last, linearly activated layer of regression models, zero may not always be an ideal choice, and may require a dedicated rule for successful application in regression scenarios. For applications of LRP in regression settings, the choice of reference activation value is well motivated and explained in (Letzgus, 2022). Choosing a suitable reference value for LRP depends on the intended explanation target. However, setting such reference during active training, might be non-trivial, as it strongly affects the magnitude and direction of parameter updates, and should be investigated in dedicated future work.
>
> We detail this in a newly added Appendix Section A.8. There, we also provide an initial experiment training a small MLP regressor on the California Housing Dataset with LFP. We utilize $r_k = (y_k-o_k)^3 \cdot \text{sign}(o_k)$ and do not change the reference value from zero. In this example, the application of LFP can be considered successful , achieving a final MSE of $0.33$ on the test set.
>
> ### Application to Transformer Models
> GeLU activations can be handled by LFP out of the box, as they are sign-preserving (e.g., cf. our experiments with LFP on SiLU in Table 3, which is similarly shaped). LayerNorm can be handled using the same rule as we used for BatchNorm (Equations 11 and 12), except with mean and variance computed across features instead of samples.
>
> In transformer models, the main challenge for LFP is propagation through the highly nonlinear softmax function in the attention. The issue is described and solved in (Achtibat, 2024), but obtaining neuron contribution (which LFP relies on) specifically for attention layers is still an active field of research. We discuss this in the newly added Appendix A.9 of the paper.
>
> There, we provide an initial experiment that applies LFP to a ViT, handling GeLU and LayerNorm as described above, and otherwise adapting the rules suggested by (Achtibat, 2024). The model is trained on a small bean disease classification dataset (https://huggingface.co/datasets/nateraw/beans). While LFP provides a meaningful training signal, it is outperformed by gradient descent in this experiment, emphasizing the need for further research.
>
> ### Citations
> *Letzgus, Simon, et al. "Toward explainable artificial intelligence for regression models: A methodological perspective." IEEE Signal Processing Magazine 39.4 (2022): 40-58.*
>
> *Reduan Achtibat, et al. “AttnLRP: Attention-aware layer-wise relevance propagation for transformers.”
> In Proceedings of the 41st International Conference on Machine Learning, ICML 2024, volume
> 235 of Proceedings of Machine Learning Research, pp. 135–168. PMLR, 21–27 Jul 2024.*

---

> ### Author Response · Authors · 2025-04-05
> **Integrated additional Related Work on Gradient-free Approaches**
>
> We have now updated the paper to include additional related work on gradient-free and biologically (more) plausible approaches.

---

> ### Author Response · Authors · 2025-04-07
> **Added additional SNN experiment, comparing convergence speed**
>
> We have updated the paper to add an additional experiment to Section 4.2 (Figure 10)  showing the performance of LFP, surrogate gradient descent, and R-STDP on a slightly deeper LeNet model. Here, LFP achieves a similar performance to R-STDP, lower than surrogate gradient descent. In terms of speed, LFP converges faster than R-STDP and slightly slower than surrogate gradient descent.
>
> Note that due to time constraints, we were not able to investigate multiple variations of hyperparameters, which, as indicated by Figures 9 and A.3, LFP can be relatively sensitive to on SNNs.

---

### Review · Reviewer_AAUM · 2025-03-24

**Summary Of Contributions:**

The paper introduces LFP, a supervised training algorithm for training layer neural networks. The claim is that LFP does not require differentiation, which allows for more diverse model architectures with non-differentiable activation functions. A proof was given for convergence when using ReLUs. Empirically, it results sparser parameters. Computationally it is equivalent to backprop.

**Audience:**

Yes

**Broader Impact Concerns:**

No ethical concerns.

**Claims And Evidence:**

No

**Requested Changes:**

1. More care in defining the update rule. Please include layer indices on all r, w, z, etc.

2. Please have a look as biologically plausible learning rules, "Assessing the Scalability of Biologically-Motivated
Deep Learning Algorithms and Architectures" is a good read. It mentions similar algorithms that you may be interested in: "Feedback alignment", "Target Propagation", and "Difference Target-Propagation".

3. Please rebut my concerns about not needing differentiation. What exactly do you mean by that and why is it worth avoiding it?

4. Please work on the story. The method does not seem to work with the only non-differentiable nonlinearity which I thought was the main point of the approach.

**Strengths And Weaknesses:**

Strengths:

1) Research into gradient-free learning algorithms is interesting from a flexibility, computational, and biologically plausibility angle.
2) Local update rules (like Hebbian learning) win Nobel prizes.

Weaknesses:

1) My main concern reading the paper is that the learning rule seemed very similar to BackProp, and indeed in Theorem 1 shows that it is equivalent up to rescaling when using ReLUs. In fact it seems carefully crafted to behave like BP when using ReLUs. When using other nonlinearities it is roughly equivalent to using backprop but ignoring the nonlinearity gradients in the chain rule (and including a term with the activations that will zero out the rewards if the activations are zero). Table 3 shows that the method usually performs best when the nonlinearity is ReLU, which makes me question its utility.

2) "No differentiation". There are error (or reward) signals being passed back through the network. In biological learning literature differentiation-free usually means avoiding the need for "weight-transport": when y_j = sum_i W_ij x_i on the forward pass, avoiding computing anything with the transpose of W, for example avoiding computing a term line sum_j W_ij r_j . However I believe you do have that term in your update when you compute r_i = sum_j r_ij. Therefore I believe there is differentiation of linear parts of the NN, so I am worried about the claim of no differentiation.

3) The paper hinted that it would work well with non-differentiable nonlinearities (mentioning heaviside as a motivating example). I was surprised in Table 3 when the method did not work at all with heaviside (and in fact backprop somehow managed to do better? was it managing to at least train the final layer?). This really confused me as I thought the main motivation was handling these types of networks.

4) Missing some citations to bio-plausible learning literation (see next section).

---

> ### Author Response · Authors · 2025-04-03
> **Clarification on “No differentiation” and Relation to Gradient Backprop**
>
> Thank you for your feedback! As you suggested, we added layer indices in a newly uploaded, revised version of the submission. We will also add additional related work on biologically plausible learning to the related work in a later version this week.
>
> ### Clarification on “No differentiation” and Relation to Gradient Backprop
> As you have mentioned, and as indicated by Theorem 1 in the paper, LFP and Gradient Backprop do share similarities; for instance, both algorithms perform backpropagation of their respective feedback signals (i.e., error for GD and reward for LFP). In fact, for specific activations $\phi$, the LFP-0-Rule, and a reward function equal to  $r_c = o_c \cdot \left(-\frac{d L}{d o_c}\right)$, the updates performed by both methods are equivalent. However, these conditions are not met
> - if rules other than LFP-0 are applied. In practice, we employ LFP-$\varepsilon$, which allows for numerical stability by regularizing the magnitude of feedback propagated backward.
> - if $\phi$ is not piece-wise linear (e.g. Tanh, GeLU, SiLU) or if $z^l_j \cdot \frac{d \phi(z^l_j)}{d z^l_j} \neq \phi(z^l_j)$ (e.g., non-differentiable functions such as Heaviside or constant functions, cf. also response to Heaviside-performance below). Note that we initially described this condition incorrectly in the paper, which we fixed and clarified in the revised version.
> - if an initial reward *not* equal to  $r_c = o_c \cdot \left(-\frac{d L}{d o_c}\right)\)$ is chosen. This may include non-differentiable rewards, such as the discrete rewards #1 and #2 from Table 2.  Although training with these two rewards is unstable in practice, we show that LFP provides a valid training signal when hyperparameters are chosen carefully (cf. Appendix A.3).
>
> We made the these conditions more clear in Section 3.3 in the revised version.
> From the perspective of gradient backprop, LFP is thus relaxing several differentiability constraints on the model and training objective, providing additional flexibility for modelling. However, it also imposes some new constraints such as sign preservation, as detailed in our response to Reviewer “LAuB”. Specifically, avoiding gradients eases the training of computationally efficient models with non-continuous forward passes, e.g. models with quantized activations or SNNs (Neftci, 2019)(Long, 2021).
>
> This property is what we refer to as “not requiring differentiation” in the paper, since computing LFP does not require the computation of gradients. However, LFP is built around the backward pass of LRP, propagating a reward. Our methodology therefore indeed differs from "derivative-free" optimization methods (Kramer et al., 2011) in a key aspect. While previous derivative-free approaches merely require the ability to evaluate an objective function $f$ at some input $x$ without assuming knowledge of the shape of the function $f$, our proposed approach is fundamentally "gradient-free" in a different sense.
>
> Unlike gradient descent methods that rely on automatic differentiation, our approach performs the reward backward pass using fixed, predetermined rules that do not necessitate explicit gradient calculations. Although we utilize network weights and activations in our computations—which could be interpreted as a form of differentiation for the linear components of the neural network—these values are directly read from the network structure without employing traditional gradient computation techniques. This approach is particularly advantageous in scenarios where gradient calculations are either highly susceptible to significant numerical errors or computationally prohibitive to derive, such as in complex, non-differentiable, or high-dimensional optimization landscapes.
>
> To make the distinction more clear, we have changed the terminology to “gradient-free” in the paper instead. In this context, we would like to mention that if Theorem 1 is not met, LFP employs a fundamentally different update strategy even on very simple models, as discussed in the newly added Appendix A.4. There, the task was to maximize/minimize a single, SiLU-activated neuron. Note that the conditions of Theorem 1 are not met by the SiLU non-linearity. In this setting, LFP finds the correct but unreachable maximum, while GD converges to the correct minimum.
>
> ### Citations
> *Neftci, Emre O., Hesham Mostafa, and Friedemann Zenke. "Surrogate gradient learning in spiking neural networks: Bringing the power of gradient-based optimization to spiking neural networks." IEEE Signal Processing Magazine 36.6 (2019): 51-63.*
>
> *Long, Ziang, Penghang Yin, and Jack Xin. "Learning quantized neural nets by coarse gradient method for nonlinear classification." Research in the Mathematical Sciences 8 (2021): 1-19.*
>
> *Kramer, Oliver, David Echeverría Ciaurri, and Slawomir Koziel. "Derivative-free optimization." Computational optimization, methods and algorithms. Berlin, Heidelberg: Springer Berlin Heidelberg, 2011. 61-83.*

---

> ### Author Response · Authors · 2025-04-03
> **Additional Analysis of LFP-Performance on Heaviside-activated Models**
>
> ### Performance of LFP on Heaviside-activated Models
> Despite being able to propagate a reward through Heaviside activations in theory, Table 3 indeed shows a negative result for LFP. This is initially surprising, especially compared to the GB-trained models. However, since the models use ImageNet pretrained weights and have a linear activation in the last layer, it is reasonable that GD will be able to score better thandoes contain random performance in these tasks, by only updating the last layer.
>
> In the paper, we hypothesize that LFP is not successful on the Heaviside-activated ANNs of that experiment due to initial updates resulting in a large number of dead (=always zero activating) neurons. Because such neurons do not contribute and consequently do not propagate rewards backward, they essentially block both the forward and backward pass and prevent lower layers from updating. Note, however, that LFP is able to successfully train the SNNs in Section 4.2, which are also Heaviside-activated. We attribute this discrepancy to the noisy, time-varying inputs of SNNs and their accumulation of potential over time.
>
> If the above hypothesis is true, (1) this increase in zero-activating neurons should be observable when training Heaviside-activated ANN with LFP (as opposed to GD, where nothing should change) and (2) Adding a small amount of noise to activations (there seems to be evidence for noisiness in biological neurons as well (Buhmann, 1987)) in the forward pass (only during training) should mitigate the “blocked” backward/forward passes for LFP, and result in meaningful (albeit slightly noisy) updates. We test this experimentally on MNIST in the newly added Section 3.6, using a small three-layer MLP which is fully Heaviside-activated (including the last layer) and a small LeNet (Heaviside-activated except for the last layer). As shown by the plots there, (1) “dead” neurons occur quickly after the first few iterations (for LFP; nothing changes for GD) and (2) adding small noise to activations mitigates this issue for LFP only, resulting in a final, Heaviside-activated model with vastly improved test accuracy (no noise applied during testing). Note that despite the last LeNet layer being linearly activated, GD only achieves random performance; we therefore attribute the above-random performance of GD in Table 3 to the pre-trained weights advantage, combined with the linearly activated (and thus GD-trainable) last layer. We have also adapted the narrative in Section 3.5 (Revised Version) accordingly. Thank you for pointing out this issue and helping us improve the paper.
>
> Note that a limitation of this noise approach seems to be considerably increased training times (300 epochs for the MLP and 400 for the LeNet, respectively). Different noise, or a more informed “drop-in” strategy may be able to resolve this, however, we leave investigation into this to future work.
>
> ### Citations
> *Buhmann, Joachim, and Klaus Schulten. "Influence of noise on the function of a “physiological” neural network." Biological cybernetics 56.5 (1987): 313-327.*

---

> ### Author Response · Authors · 2025-04-05
> **Integrated Requested Changes to Related Work**
>
> We have now updated the paper to include additional related work on biologically plausible learning.

---

### Decision · Action_Editor_Bs4B · 2025-05-07

**Recommendation:** Accept with minor revision

**Comment:**

Overall the reviewers found the method interesting, and the work well-written. Concerns raised by the reviewers rested in the potential similarities to backpropagation in restricted cases, and the "no differentiation" terminology used to describe the method given the methodology. Further concerns included the substantial restrictions/limitations of the method e.g. activation functions, and missing literature in both the bio-plausible and gradient-free training approaches. The authors responded to almost all of the reviewer's concerns effectively and did incorporate some of those changes into the manuscript during the rebuttal itself to address the concerns.

Post-rebuttal, all reviewers found their concerns adequately addressed, with the sole exception of some questions remaining on the scalability of the method. Reviewer recommendations post-rebuttal noted that while the methodology itself might be counter-intuitive, has significant limitations, and imposes significant restrictions, all of these limitations/restrictions were clearly discussed by the authors in the paper. Given this, the work was judged to be within the TMLR acceptance criteria.

While I'm aware many of the changes requested by reviewers during the rebuttal period were already updated in the manuscript, due to the large number of changes and >12 page length of the paper I have chosen to accept with a minor revision to ensure all the requested changes are completed before acceptance, and also in order to allow the authors to further address scalability of the method in particular on deeper/closer to state-of-the-art architectures.

**Audience:**

While some reviewers worried that the significant limitations/restrictions of the method as outlined by the authors and potential lack of scalability would curtail the interest in the work, there was enough interest in the method as outlined by the reviewers on a whole that I am convinced the work would likely be of interest to some in TMLR's audience. In particular one reviewer thought the novelty in the method could even be of interest to those in the broader ML community.

**Claims And Evidence:**

The reviewers consistently found there was sufficient evidence to support the claims made by the authors, and just as importantly, that the restrictions/limitations of the method were clearly detailed in the paper.